# Optogenetic cleavage of the Miro GTPase reveals the direct consequences of real-time loss of function in *Drosophila*

Francesca Mattedi, Ethlyn Lloyd-Morris, Frank Hirth, Alessio Vagnoni◉*

Department of Basic and Clinical Neurosciences, Maurice Wohl Clinical Neuroscience Institute, Institute of Psychiatry, Psychology and Neuroscience, King's College London, London, United Kingdom

* alessio.vagnoni@kcl.ac.uk

## Abstract

Miro GTPases control mitochondrial morphology, calcium homeostasis, and regulate mitochondrial distribution by mediating their attachment to the kinesin and dynein motor complex. It is not clear, however, how Miro proteins spatially and temporally integrate their function as acute disruption of protein function has not been performed. To address this issue, we have developed an optogenetic loss of function "Split-Miro" allele for precise control of Miro-dependent mitochondrial functions in *Drosophila*. Rapid optogenetic cleavage of Split-Miro leads to a striking rearrangement of the mitochondrial network, which is mediated by mitochondrial interaction with the microtubules. Unexpectedly, this treatment did not impact the ability of mitochondria to buffer calcium or their association with the endoplasmic reticulum. While Split-Miro overexpression is sufficient to augment mitochondrial motility, sustained photocleavage shows that Split-Miro is surprisingly dispensable to maintain elevated mitochondrial processivity. In adult fly neurons in vivo, Split-Miro photocleavage affects both mitochondrial trafficking and neuronal activity. Furthermore, functional replacement of endogenous Miro with Split-Miro identifies its essential role in the regulation of locomotor activity in adult flies, demonstrating the feasibility of tuning animal behaviour by real-time loss of protein function.

## Introduction

Methods to observe loss of function (LoF) phenotypes are used to study many biological processes. Although important tools for elucidating gene function, disruption of genes by genomic mutations or RNA interference (RNAi) often does not have the spatiotemporal resolution to capture the direct cellular and organismal consequences of LoF and to report specifically on a protein's primary function. The observed "end-point" phenotypes might thus be the result of compensatory mechanisms to loss of protein, and gene pleiotropy means that, even in cell culture models, it is often difficult to dissect the causality of the observed phenotypes.

This issue is especially problematic when studying complex cellular processes such as mitochondrial dynamics, and it is particularly evident for Miro proteins, mitochondrial Rho

**Data Availability Statement:** All relevant data are within the paper and its Supporting information files.

**Funding:** This work was supported by a NC3Rs David Sainsbury fellowship (NC/N001753/2) and NC3Rs SKT grant (NC/T001224/1), an Academy of Medical Sciences Springboard Award (SBF004/1088), an ARUK King's College London Network Centre Grant (ARUK-NC2020-KCL), a van Geest Fellowship in Dementia and Neurodegeneration, and a van Geest Studentship to A.V. A MRC-DTP Studentship supports E.L.M. The funders had no role in study design, data collection and analysis, decision to publish, or preparation of the manuscript.

**Competing interests:** The authors have declared that no competing interests exist.

**Abbreviations:** AR, aspect ratio; DART, *Drosophila* ARousal Tracking; fps, frame per seconds; HRP, horseradish peroxidase; LoF, loss of function; MERCS, mitochondria-ER contacts site; RNAi, RNA interference; ROI, region of interest; wt-Miro, wild-type Miro.

GTPases that influence the motility, morphology, and physiology of mitochondria [1–3]. In *Drosophila*, homozygous mutants of *miro* are developmentally lethal [4], while knockout of *Miro1* in mice leads to perinatal lethality [5,6]. Chronic loss of Miro is detrimental for mitochondrial transport in *Drosophila* and mammalian neurons [7–9], where it leads to alteration of synaptic strength [4,10], whereas disruption of Miro in *Drosophila* additionally impairs mitochondrial calcium homeostasis [11,12]. Reducing Miro abundance has also profound effects on mitochondrial morphology with fragmented mitochondria observed in yeast [13], *Drosophila* larval motor neurons [7], and mouse embryonic fibroblasts [6]. It is not yet clear, however, to what extent these phenotypes are directly consequences of Miro disruption and whether they might arise independently. Any attempt to dissect the causality of cellular and organismal phenotypes after Miro manipulation is challenging as the current tools do not allow acute disruption of function.

Here, we present the generation of "Split-Miro," a photocleavable variant of *Drosophila* Miro, to achieve rapid and controlled loss of protein function. Mitochondrial network remodelling is a rapid response to Split-Miro photocleavage in *Drosophila* S2R+ cells. This effect, mediated by loss of anchorage to the microtubule network, impacts on mitochondrial distribution into cell processes where the use of Split-Miro shows that this protein is sufficient to increase mitochondrial motility but dispensable for the maintenance of elevated mitochondrial velocities. This effect is mirrored in adult fly neurons in vivo, where Split-Miro affects the motility, but not the processivity, of the mitochondria. Unexpectedly, we show that Split-Miro photocleavage neither directly impacts mitochondrial calcium homeostasis nor the association between the mitochondria and the endoplasmic reticulum in S2R+ cells. Finally, we demonstrate that Split-Miro modulates neuronal activity in adult flies, and, by rescuing the lethality associated with classical Miro LoF mutations via pan-neuronal expression, we provide proof of concept that Split-Miro affords control of fly locomotor activity through exposure to blue light.

## Results

### Design of a photocleavable Miro variant in *Drosophila*

To gain real-time spatiotemporal control of Miro LoF, we created a Miro variant that contains the LOV2-Zdk1 protein pair (Fig 1A and 1B) that undergoes light-induced dissociation upon exposure to blue light [14–16] and so is predicted to achieve rapid and reversible Miro LoF through protein photocleavage (Fig 1B).

We fused Zdk1 N-terminally to the mitochondrial targeting transmembrane domain of Miro, with LOV2 fused C-terminally to the rest of the protein (Fig 1A and 1B), and expressed this optogenetic variant of Miro (herein called "Split-Miro") as 2 components in *Drosophila* S2R+ cells (Figs 1B, 1C, and S1A). In absence of blue light, LOV2 interacts with Zdk1, thus reconstituting Split-Miro at the mitochondria (Fig 1C). However, upon blue light exposure, the LOV2 Jα helix undergoes a conformational change that prevents Zdk1 from binding [14,15], resulting in the photocleavage of Split-Miro and the complete and rapid release of the N-terminal moiety into the cytoplasm (Fig 1D–1F and S1 Movie). Quantifications of the kinetics of release after blue light exposure showed release half-life of 1.6 ± 0.2 seconds (Fig 1F) and full Split-Miro reconstitution within 3 minutes (Fig 1D, 1E and 1G) after removal of blue light, in line with what was previously reported for the LOV2-Zdk1 association in mammalian cells [14,17]. Tagging Split-Miro only at the N-terminus did not significantly change the kinetic of photocleavage (S1B–S1G Fig and S2 Movie), thus showing the Split-Miro design supports a versatile tagging strategy to facilitate fluorophore multiplexing.

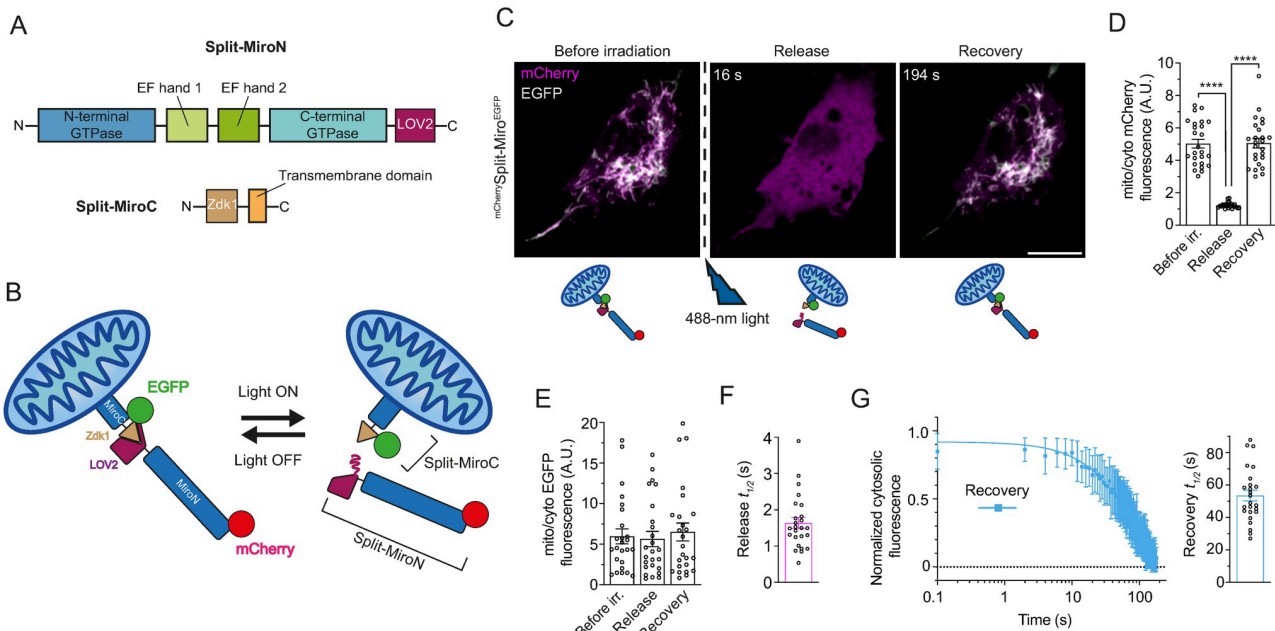

**Fig 1. Split-Miro is a photocleavable version of *Drosophila* Miro. (A)** Schematic of Split-Miro functional domains. **(B)** Schematic representation of reconstituted and photocleaved mitochondrial-bound Split-Miro in its dark and lit state, respectively. Split-MiroN and Split-MiroC moieties can be followed independently by tagging them with different combinations of fluorescent proteins (e.g., EGFP/mCherry). **(C)** Localisation of Split-Miro in S2R+ cells cotransfected with mCherry-Split-MiroN (magenta) harbouring the T406A, T407A mutations in the N-terminus of LOV2 [15,18], and EGFP-tagged Split-MiroC, before and after blue light exposure. Before irradiation, the mCherry and EGFP signals colocalise, while mCherry-Split-MiroN is fully released into the cytoplasm (indicated by the homogenous magenta colour) immediately after irradiation (16 seconds), and it fully reconstitutes after 3 minutes. Cartoon depicts photocleavage and reconstitution of mCherry-Split-Miro-EGFP at different time points. s, seconds. **(D)** Quantification of mCherry-Split-MiroN fluorescence at the mitochondria and in the cytoplasm before irradiation, at the point of maximum release and recovery indicates that the N-terminus moiety is completely released into the cytoplasm (mean ratio = 1 at release) and fully reconstituted afterwards. **(E)** Quantification of EGFP-Split-MiroC fluorescence indicates that the localisation of the C-terminus moiety does not change during the experiment. **(F, G)** Quantification of mCherry-Split-MiroN half-time release (F) and recovery (G) after photocleavage, relative to C. In (G), left panel shows levels of cytosolic Split-Miro N-terminus quantified after the maximum release is reached; right panel: recovery half-time. Data are shown as mean ± SEM. Solid line in (G) is exponential curve fit. Circles, number of cells, from 3 independent experiments. Statistical significance in (D, E) was calculated with a repeated measures one-way ANOVA followed by Tukey's post hoc test. Scale bars: 10 μm. **** $p < 0.0001$. The data underlying the graphs shown in the figures can be found in S1 Data.

## Overexpression of Split-Miro increases the proportion of motile mitochondria and their processivity in cell processes

*Drosophila* S2R+ cells are often used for intracellular trafficking experiments [19,20] and can be induced to extend processes (Fig 2A) with a stereotypical plus-end out microtubule array [20,21] that display Miro-dependent long-range, bidirectional mitochondrial motility (Figs 2B–2D and S2A–S2C). With the aim of studying whether mitochondrial motility can be manipulated via Split-Miro, we first transfected S2R+ cells with either Split-Miro or wild-type Miro (wt-Miro) N-terminally tagged with mCherry. We initially exploited the mCherry tag to follow mitochondria with a 561-nm laser line, which does not photocleave Split-Miro (Fig 2E–2G; 561-nm laser).

Miro links kinesin and dynein motors to mitochondria via milton (TRAK1/2 in mammals) [8,9,22–25] (S2D and S2E Fig), and so overexpression of Miro is predicted to favour the recruitment of motor proteins on mitochondria for processive transport. Consistent with this idea, mitochondria spent more time on long runs, paused less, and engaged less frequently in short runs after either Miro or Split-Miro overexpression compared to controls (Fig 2F, 561-nm laser, and S2F and S2G Fig). Thus, both Miro isoforms turn the motility of

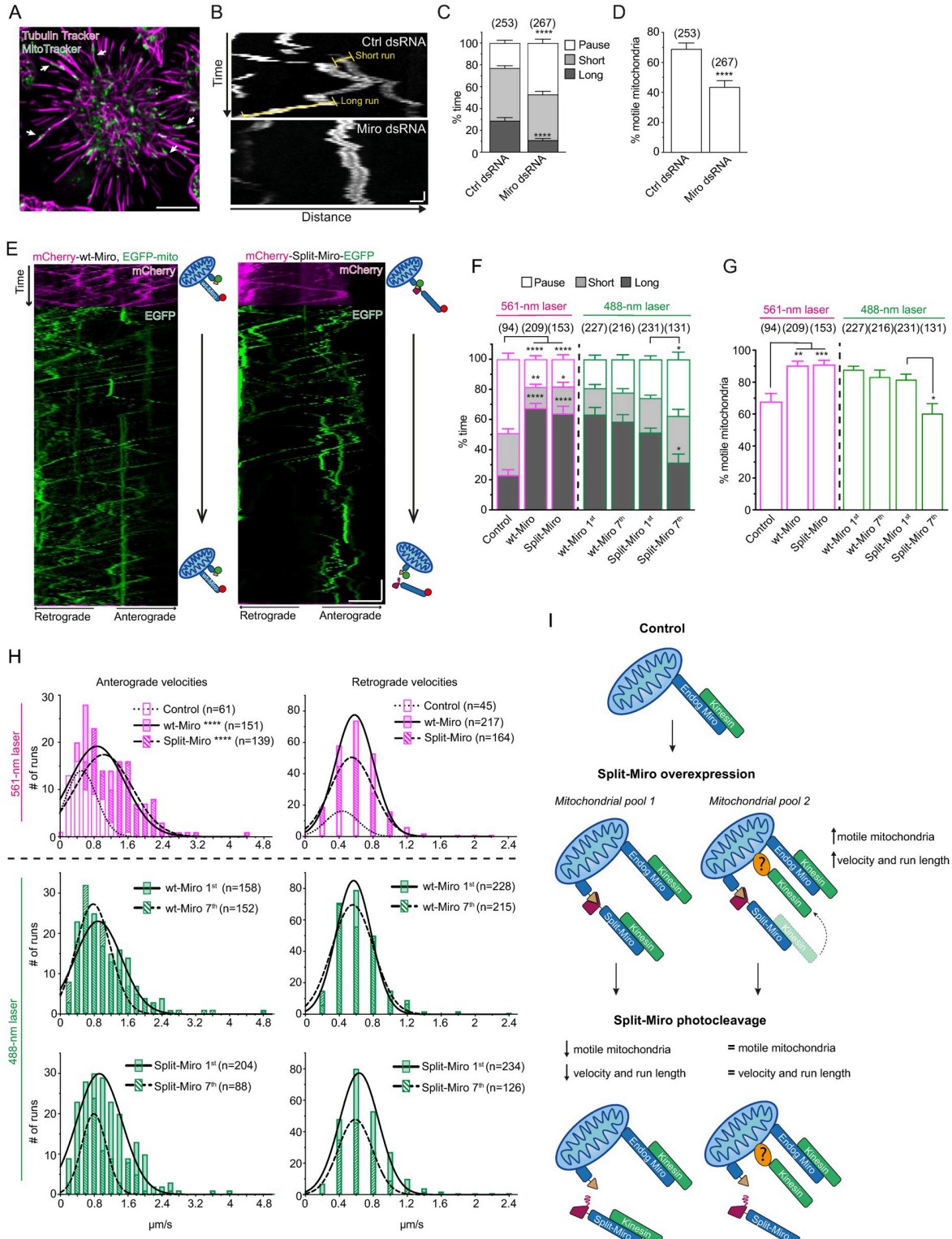

**Fig 2. Miro regulates mitochondrial motility in the processes of S2R+ cells, and Split-Miro photocleavage reverts the effect of Split-Miro overexpression on mitochondrial motility but not velocity. (A)** Example of S2R+ cell treated with cytochalasin D to induce the formation of microtubule-rich processes. The microtubules and the mitochondria are stained with Tubulin Tracker and with MitoTracker Green (MitoTracker), respectively. White arrows indicate examples of cellular processes containing mitochondria. Scale bar: 10 μm. **(B)** Representative kymographs of mitochondrial transport in the cellular processes of S2R+ cells treated with control (upper panel) or Miro

dsRNA (bottom panel). Yellow highlights indicate examples of short (<2 μm) and long runs (≥2 μm). Scale bars: 1 μm (distance) and 5 seconds (time). **(C)** Duty cycle analysis describes the average time mitochondria spend on long runs, short runs, or pausing. For each parameter, all mitochondrial values from each cell were averaged and compared between control and Miro dsRNA condition. **(D)** Percentage of motile mitochondria in cellular processes. Number of mitochondria analysed are in brackets, from 29 (Ctrl dsRNA) and 36 (Miro dsRNA) cells, from 2 independent experiments. Data are shown as mean ± SEM, multiple Student's *t* tests (C) and Mann–Whitney test (D). This analysis suggests that Miro is necessary to drive mitochondrial motility in *Drosophila* S2R+ cells, and this is mainly due to modulation of long-range transport, while short-range motility appears to be largely independent of Miro. **(E)** Representative kymographs showing mitochondrial transport in S2R+ cellular processes before and during exposure to blue light. Control cells were cotransfected with mCherry-tagged wt-Miro, which cannot be photocleaved, and EGFP targeted to the mitochondria via the Zdk1-MiroC anchor (EGFP-mito). Split-Miro was tagged with both EGFP and mCherry to independently follow the C-terminal and N-terminal half, respectively. Mitochondrial transport was first imaged with a 561-nm laser, to capture the mCherry signal, and then with a 488-nm laser, to capture the EGFP signal while photocleaving Split-Miro (S4 and S5 Movies). Cartoon depicts schematic of the transfected constructs. Scale bars: 5 μm (distance) and 30 seconds (time). **(F)** Duty cycles analysis and **(G)** percentage of motile mitochondria in cellular processes. In (F, G), mitochondrial transport was first analysed following the mCherry tag in cells expressing either mCherry-tagged wt-Miro or Split-Miro (561-nm laser). In a separate experiment (488-nm laser), mitochondrial transport was quantified following the EGFP tag at the beginning (first minute) and at the end (seventh minute) of the time-lapse imaging with 488-nm blue light, as shown in panel E. In (F), for each parameter, all mitochondrial values from each cell were averaged and compared to the control condition (561-nm laser) or to the first minute of blue light exposure (488-nm laser). Data are shown as mean ± SEM, from 3 independent experiments. Statistical significance was evaluated by one-way ANOVA followed by Tukey's post hoc test (561-nm laser) or multiple paired *t* tests (488-nm laser) in (F) and by Kruskal–Wallis test followed by Dunn's post hoc test (561-nm laser) or by Mann–Whitney tests (488-nm laser) in (G). Number of mitochondria analysed are in brackets, from 16 (Control), 15 (wt-Miro), and 15 (Split-Miro) cells under 561-nm laser, and 11 (wt-Miro) and 17 (Split-Miro) cells under 488-nm laser. **(H)** Distribution of anterograde and retrograde long run velocity after wt-Miro and Split-Miro overexpression (561-nm laser) and after irradiation with blue light to photocleave Split-Miro (488-nm laser). Solid, dashed, and dotted lines are fitted curves. Statistical significance was calculated with a Kruskal–Wallis test followed by Dunn's post hoc test (561-nm laser) and a Mann–Whitney test (488-nm laser). *N* = number of mitochondrial runs. * $p < 0.05$, ** $p < 0.01$, *** $p < 0.001$, **** $p < 0.0001$. **(I)** Model for Split-Miro–mediated regulation of mitochondrial motility. The Split-Miro trafficking data are consistent with a model in which 2 different pools of mitochondria coexist in S2R+ cells. After overexpression, Split-Miro recruits the motor complexes on the mitochondria and activates transport. On one mitochondrial pool (*mitochondrial pool 1*), Split-Miro links directly or indirectly (for example, via Milton/Trak, not depicted) to the motor proteins. On another subset of mitochondria (*mitochondrial pool 2*), the motor complexes are instead stabilised on the organelle by an unknown factor (question mark) following Miro-dependent recruitment. After Split-Miro photocleavage, *mitochondrial pool 1* can reverse to control levels of processivity or become stationary (Fig 2F and 2G). However, the processivity (e.g., velocity and run length) of the *mitochondrial pool 2* is not affected as the recruited motors are not directly linked to the mitochondria via Miro (Figs 2H, S2J and S2K). This model is consistent with previous reports describing Miro-independent mitochondrial motility [6,24]. Split-Miro–dependent recruitment of motor complexes on an interconnected mitochondrial network, such as the one found in the perinuclear area of the cell soma, could cause significant tension on the network. Releasing the Split-Miro anchor by photocleavage, even if only on a subset of the mitochondria, would be sufficient to release the tension and cause mitochondrial network retraction, as shown in Fig 3. Not depicted, dynein motor complex. The data underlying the graphs shown in the figures can be found in S1 Data.

mitochondria from predominantly bidirectional with frequent reversals to markedly more processive. Further supporting this notion, both Miro isoforms caused a higher proportion of mitochondria in cells processes to be motile (Fig 2G, 561-nm laser), and their duty cycle increased in both the anterograde and retrograde directions (S2H Fig), suggesting that Miro participates in the activation of transport complexes for bidirectional mitochondrial transport. Interestingly, while there was no difference in the retrograde run velocities when Miro or Split-Miro were overexpressed, mitochondria traveling in the anterograde direction moved at nearly double the speed when compared to control (Fig 2H, 561-nm laser). This observation might reflect a preference for Miro to recruit milton-kinesin complexes, consistent with what was observed in larval segmental neurons after Miro overexpression [26]. Collectively, these experiments shed light on Miro's role in regulating mitochondrial motility in S2R+ cells and demonstrate that, in absence of blue light exposure, Split-Miro and Miro are functionally equivalent.

### Optogenetic cleavage of Split-Miro reduces the proportion, but not the processivity, of motile mitochondria in the cell processes

To test whether Split-Miro photocleavage reverses the observed Miro gain-of-function effects on mitochondrial transport, we analysed mitochondrial motility under blue light in wt-Miro

and Split-Miro transfected cells. Time-lapse imaging with the 488-nm laser line ensured that Split-Miro was not reconstituted while recording mitochondrial motility via the EGFP-tag (Fig 2E, green). wt-Miro, which cannot be photocleaved, was used as a control. We found that while there was no detectable difference in the motility after imaging for 1 minute (S2I Fig), sustained Split-Miro photocleavage (7 minutes) reduced the time mitochondria spent on long runs and the proportion of organelles in the processes that were motile to the levels observed prior to overexpression, while there was no such effect in wt-Miro controls (Fig 2E–2G, 488-nm laser). Cleaving Split-Miro from mitochondria did not, however, have any major effects on the velocity and run length of the moving organelles, which remained elevated and did not return to control levels (Fig 2H, 488-nm laser, and S2J Fig). This was also the case when endogenous Miro was depleted in the Split-Miro condition by RNAi (S2K Fig). Thus, using a strategy combining up-regulation (via overexpression) and down-regulation (via Split-Miro photocleavage), these results indicate that the proportion and processivity of motile mitochondria are controlled by separate Miro-dependent and Miro-independent mechanisms (Fig 2I).

## Split-Miro photocleavage triggers a rapid collapse of the mitochondrial network, which is rescued by mitochondrial anchoring to the microtubules

We next examined the effect of Split-Miro on the integrity of the mitochondrial network in the cell soma. Strikingly, exposing Split-Miro-transfected S2R+ cells to blue light triggered a rapid (<3 minutes) and dramatic remodelling of the entire mitochondrial network, which progressively collapsed towards the centre of the cell (Fig 3A and S3 Movie). Mitochondria shortened along their long axis taking up a rounder shape (Fig 3B), which was associated with a strong reduction in the number of branches (Fig 3C) and reduction of the total area covered by mitochondria (S3A Fig). Again, the presence of endogenous Miro was dispensable for Split-Miro functionality as reducing Miro levels by RNAi did not affect the mitochondrial phenotype in Split-Miro–transfected cells (Fig 3D). This phenotype is not dependent on adverse effects caused by increased cytoplasmic concentration of Split-MiroN, as overexpression of the Miro N-terminal moiety alone, which diffuses throughout the cytoplasm, does not cause any gross mitochondrial morphological aberration (S3B Fig and [24,25]). Of note, neither the motility and distribution of peroxisomes in the cell processes (S3C and S3D Fig) nor the overall organisation of the microtubule network (S3E Fig) were affected by this rapid cellular-scale change. Nevertheless, we reasoned that mitochondrial network collapse might impede organelle delivery towards the periphery. The number of mitochondria in the cell processes showed a progressive decline in mitochondrial content in cells expressing Split-Miro under blue light, while no effect was observed in the presence of wt-Miro (S3F and S3G Fig). We conclude that the rapid mitochondrial shape transition with associated loss of network integrity, clearly detectable within 3 minutes under blue light, strongly contributes to the progressive depletion of mitochondria from the cell processes.

The Miro-milton-motor complex provides a link for the attachment of mitochondria onto the microtubules [27,28]. To test the hypothesis that loss of microtubule tethering is responsible for the collapse of the mitochondrial network, we set out to induce mitochondrial tethering to the microtubules in a Miro-independent manner. In mammals, syntaphilin (SNPH) anchors mitochondria onto the microtubules [29], although a *Drosophila* homologue has not yet been found. Thus, we coexpressed EGFP-tagged human SNPH with Split-Miro in S2R+ cells stained with MitoTracker. SNPH signal in *Drosophila* cells overlaps with mitochondria and, as observed in mammalian neurons, SNPH puncta often localise at mitochondrial ends

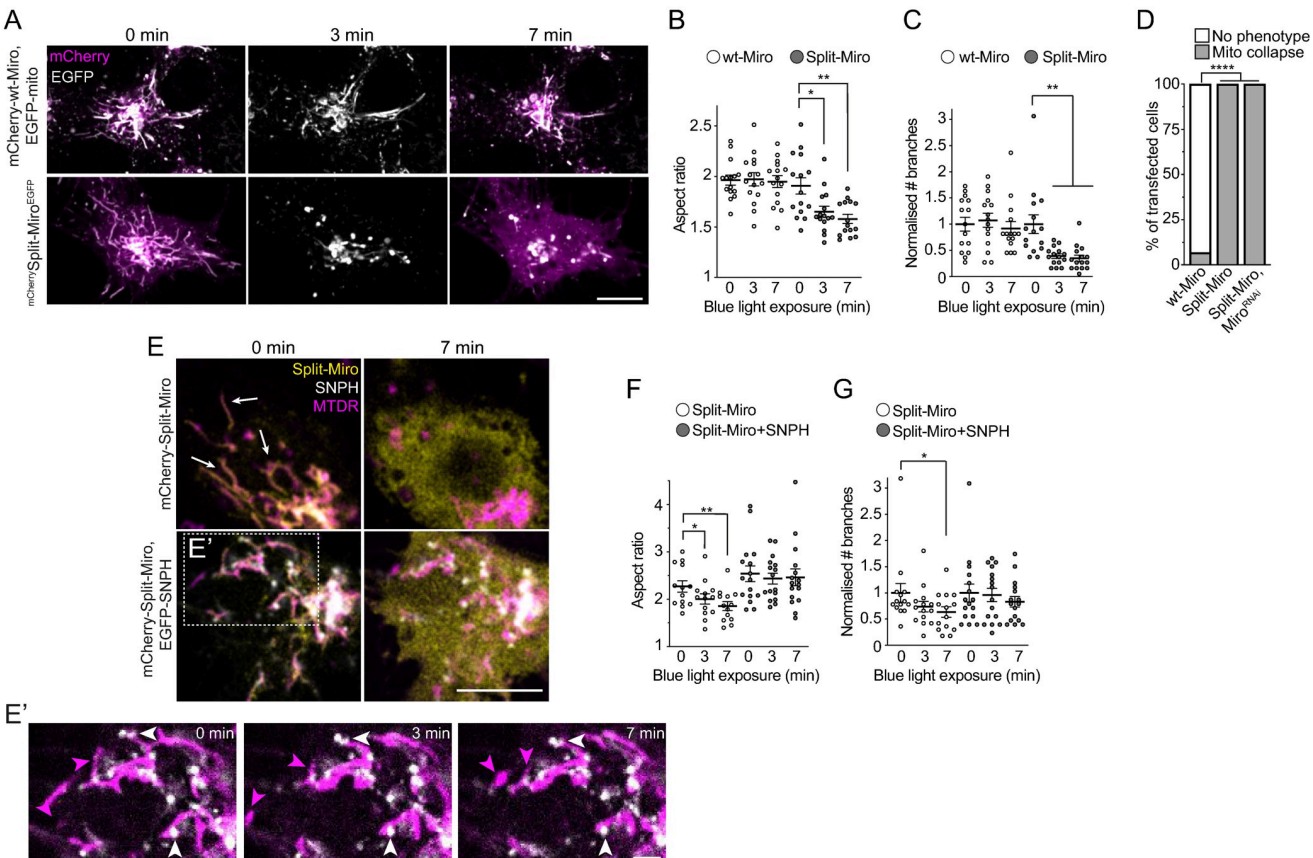

**Fig 3. Split-Miro–dependent changes of mitochondrial morphology and connectivity are rescued by SNPH expression. (A)** Representative images of mitochondria in *Drosophila* S2R+ cells at the beginning (0 minutes) and after 3 and 7 minutes of exposure to blue light, which leads to Split-Miro, but not wt-Miro, photocleavage. Control cells (top panels) were cotransfected with mCherry-tagged wt-Miro (magenta) and EGFP targeted to the mitochondria via the Zdk1-MiroC anchor (EGFP-mito, grey). Split-Miro (bottom panels) was tagged with both EGFP and mCherry to independently follow the C-terminal and N-terminal half, respectively. The mCherry is shown at the beginning and end of the imaging period to confirm retention on and release from the mitochondria in wt-Miro and Split-Miro transfected cells, respectively, under blue light. Scale bar: 10 μm. **(B)** Quantification of mitochondrial aspect ratio (AR) and **(C)** of the number of mitochondrial branches within the network, relative to A. **(D)** Quantifications of the mitochondrial collapse phenotype after 7 minutes of time-lapse imaging with 488-nm blue light in S2R+ cells overexpressing wt-Miro and Split-Miro with or without a Miro dsRNA construct (Miro$^{RNAi}$). Number of cells: wt-Miro = 15, Split-Miro = 15, Split-Miro + Miro$^{RNAi}$ = 10, Fisher's exact test. **(E)** Representative images of cells expressing mCherry-Split-Miro (Split-Miro, yellow) with either an empty vector (top panels) or with EGFP-SNPH (SNPH, bottom panels, grey) and stained with MitoTracker DeepRed (MTDR, magenta). White arrows show examples of mitochondria that have retracted after Split-Miro photocleavage. Diffuse cytoplasmic yellow signal indicates release of Split-Miro N-terminus from the mitochondria. Scale bar: 10 μm. **(E')** Magnified inset shows examples of stable SNPH-positive mitochondria (white arrowheads) and dynamic mitochondrial membranes devoid of SNPH (magenta arrowheads). Scale bar: 2 μm. Not shown, Split-Miro. **(F)** Quantification of mitochondrial AR and **(G)** number of mitochondrial branches at the time points indicated, relative to E. Circles represent the average AR calculated from single mitochondria within the same cell (B, F) and the average number of branches per cell normalised to the average group value (Split-Miro, Split-Miro + SNPH) at time point 0 (C, G). Comparison across time points was performed by repeated measures one-way ANOVA followed by Tukey's post hoc test (B, C, F) and Friedman test followed by Dunn's post hoc test (G), from 3 independent experiments. Data are reported as mean ± SEM. * $p < 0.05$, ** $p < 0.01$, **** $p < 0.0001$. The data underlying the graphs shown in the figures can be found in S1 Data.

and are associated with strong reduction in mitochondrial dynamics (Figs 3E–3E', S3H and S3I). While in cells devoid of SNPH the mitochondrial network retracts after Split-Miro photocleavage, the presence of SNPH prevents this phenotype (Fig 3E–3G). These observations support the notion that loss of mitochondrial anchoring on microtubules is responsible for the rapid mitochondrial network collapse when Split-Miro is cleaved.

## Split-Miro photocleavage does not affect mitochondrial calcium buffering or mitochondria-ER association

Mitochondria buffer calcium to help maintain cellular homeostasis and loss of Miro reduces calcium levels in the mitochondria of the *Drosophila* brain [11,12,30]. However, the mechanisms underlying decreased calcium uptake when Miro is disrupted are not understood. In S2R+ cells, Split-Miro photocleavage did not induce any changes in the fluorescent intensity of the mitochondrial calcium $[Ca^{2+}]_m$ indicator mito-GCaMP6f when compared to control cells (Fig 4A and 4B), indicating that the steady-state level of $[Ca^{2+}]_m$ is not affected by this manipulation. Optogenetic inactivation of Split-Miro also did not affect $[Ca^{2+}]_m$ uptake when S2R+ cells were challenged with ionomycin, an ionophore that causes a sharp increase in cytosolic calcium [27,31] (Fig 4C–4E). This result indicates that the mitochondrial morphological changes induced by Split-Miro are not sufficient to alter $[Ca^{2+}]_m$ homeostasis. Likewise, overexpression of SNPH, which rescues the Split-Miro–induced mitochondrial collapse, did not have any effect on $[Ca^{2+}]_m$ uptake when Split-Miro was cleaved (Fig 4C–4E).

It has been shown that a subset of Miro is found at the mitochondria-ER interface to regulate the contacts between these 2 organelles [12,32,33], and the mitochondria-ER contacts sites (MERCS) are known to mediate $Ca^{2+}$ exchange between the 2 organelles [34]. However, we did not detect any significant difference in the number of MERCS visualised using a split-GFP assay [35,36] after Split-Miro photocleavage (Fig 4F–4H). Furthermore, the morphological changes displayed by mitochondria after Split-Miro photocleavage were not associated with detectable changes in their membrane potential (S4 Fig), implying that mitochondria do not become dysfunctional during this rapid morphological transition. Together, these findings suggest that acutely modulating mitochondrial network integrity via Split-Miro/SNPH or blocking mitochondrial motility via SNPH (S3F and S3G Fig) are not sufficient to perturb $[Ca^{2+}]_m$ homeostasis in this context.

## In vivo regulation of mitochondrial motility by Split-Miro

To gain insight into the in vivo roles of Split-Miro in regulating mitochondrial dynamics, we expressed *UAS-mCherry-MiroN-LOV2* and *UAS-EGFP-Zdk1-MiroC* (herein "*UAS-Split-Miro*") with the *Appl-Gal4* driver and imaged mitochondrial motility in the axons of the adult wing neurons in situ [37]. Split-Miro is reconstituted in adult fly neurons and can be photocleaved efficiently through exposure to blue light (S5A–S5D Fig). Overexpression of Miro in *Drosophila* neurons in vivo has been shown to be detrimental for mitochondrial motility and overall neuronal viability [7,20]. However, expressing Split-Miro and the wt-Miro control in a *miro*$^{+/−}$ background is permissive towards mitochondrial motility (Figs 5A, 5B, S5E and S5F and S6 Movie). Tracking of mitochondrial movements revealed that, while exposing the *UAS-wt-Miro*$^+$ neurons to pulses of blue laser light did not cause any change in the number of motile mitochondria in 2-day-old flies (Fig 5A and 5C and S6 Movie), *UAS-Split-Miro*$^+$ neurons showed a strong decline after 5 minutes (Fig 5B and 5D). Mitochondrial velocity and run length, however, were not affected by this manipulation (Fig 5E and 5F), similarly to what observed in S2R+ cells after Split-Miro photocleavage. Interestingly, 7-day-old *UAS-Split-Miro*$^+$ neurons did not show any changes in mitochondrial motility under the same experimental conditions (S5E–S5H Fig). This suggests that in older flies, when the rates of mitochondrial transport are known to decline [20,38], alternative mechanisms might become predominant towards controlling mitochondrial trafficking.

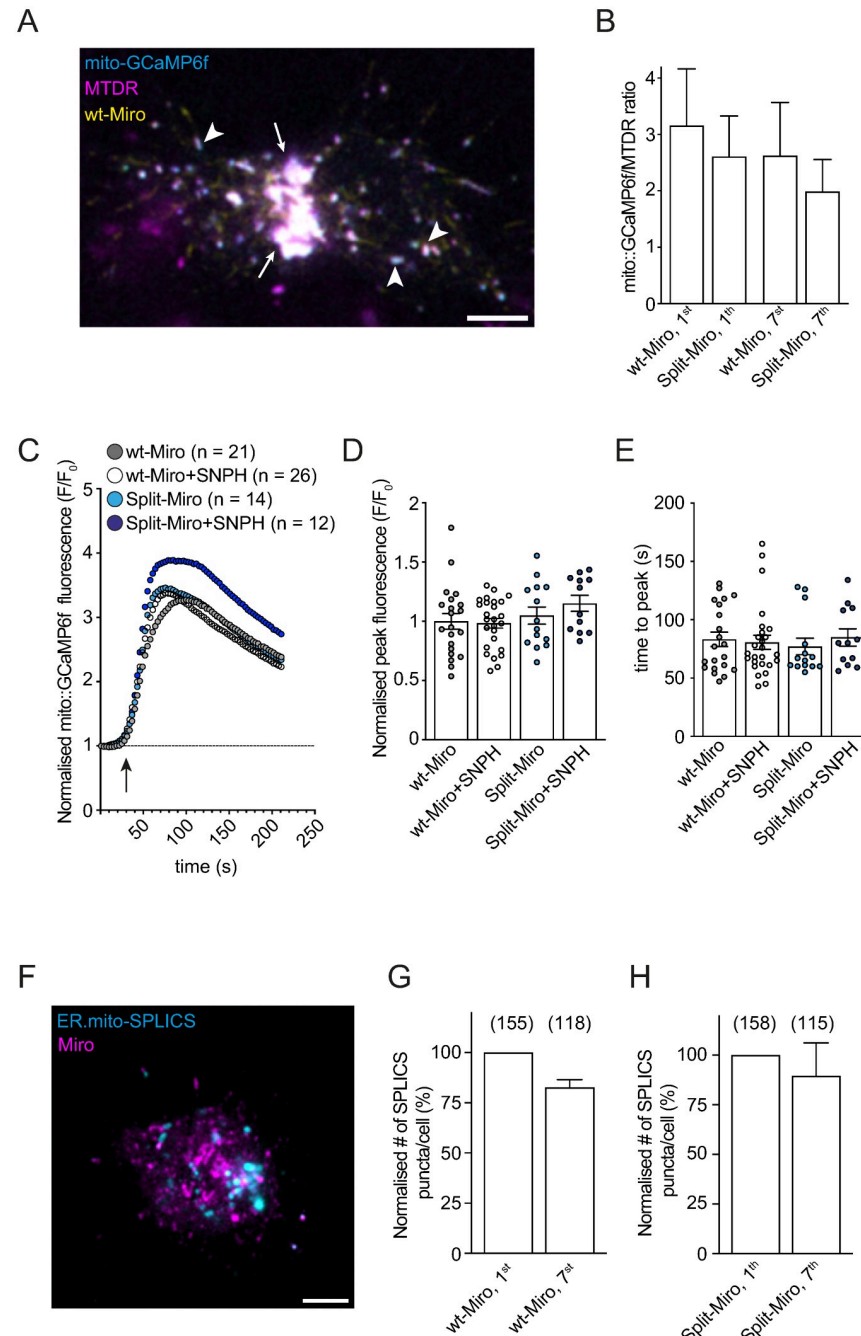

**Fig 4. Split-Miro photocleavage does not affect $[Ca^{2+}]_m$ abundance. (A)** Representative image of an S2R+ cell showing mitochondrial targeting of mito-GCaMP6f (cyan) in the perinuclear region (white arrows) and in single mitochondria (arrowheads). Yellow is mCherry-tagged wt-Miro (wt-Miro); magenta is MitoTracker DeepRed (MTDR). Scale bar: 5 μm. **(B)** Cells transfected with mito-GCaMP6f and either mCherry-wt-Miro (wt-Miro) or mCherry-Split-Miro (Split-Miro) were stained with MTDR; and the ratio of mito-GCaMP6f/MTDR signal intensity was analysed at the beginning (first minute) and at the end (seventh minute) of the time-lapse imaging under blue light. Number of cells: wt-Miro = 12, Split-Miro = 16; from 3 independent experiments. Data are shown as mean ± SEM. Repeated measures one-way ANOVA followed by Tukey's post hoc test did not show any significant difference between groups, indicating that the basal $[Ca^{2+}]_m$ does not significantly change after Split-Miro photocleavage. **(C)** Expression of mCherry-Split-Miro (Split-Miro) or EBFP-SNPH (SNPH) does not significantly alter $[Ca^{2+}]_m$ uptake in cells challenged with ionomycin, compared to control conditions. Traces indicate the average mito-GCaMP6f fluorescence intensity values (circles) at individual time point before and after cell exposure to ionomycin (arrow). *N* = number of cells, from 5 independent experiments. **(D, E)** Normalised response peak and time to reach the

peak, respectively, relative to the data shown in (C). Circles, number of cells. Kruskal–Wallis test followed by Dunn's multiple comparisons showed no difference between conditions. **(F)** wt-Miro or Split-Miro (magenta) were coexpressed with the ER-mito::SPLICS probe (cyan) in S2R+ cells. The SPLICS probe displays a typical punctuated stain in these cells, as previously observed in mammalian and *Drosophila* cells [34,36]. **(G, H)** Quantification of the SPLICS puncta and statistical analysis by Wilcoxon test showed no significant difference between the beginning (first minute) and the end (seventh minute) of the time-lapse imaging under blue light in wild-type and Split-Miro–transfected cells. Data are presented as % change of SPLICS puncta at seventh minute compared to first minute. Number of contacts analysed are in brackets from 5 (wt-Miro) and 9 (Split-Miro) cells from 2 independent experiments. The data underlying the graphs shown in the figures can be found in S1 Data.

## Split-Miro controls neuronal activity in vivo

Having observed a strong reduction in mitochondrial trafficking in 2-day-old Split-Miro flies, we next asked whether rapid Split-Miro photocleavage could affect wider neuronal physiology. Reduced mitochondrial motility has been associated with alterations in neuronal activity [4,10,39,40]. Thus, we decided to measure the activity of adult wing neurons by recording

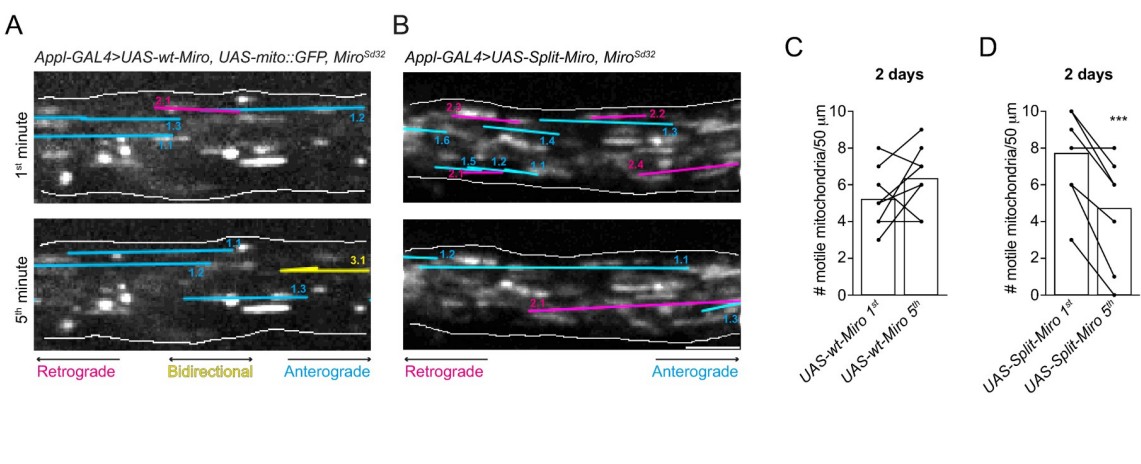

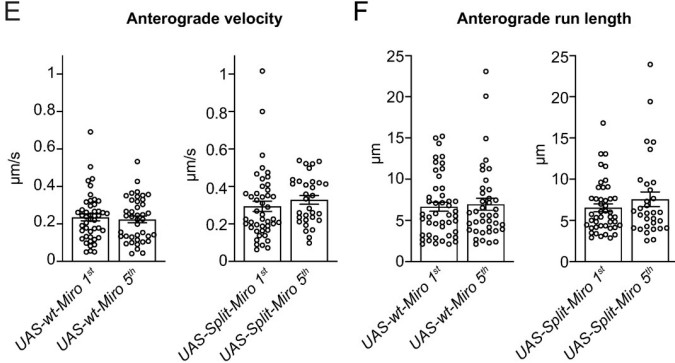

**Fig 5. Split-Miro–dependent control of mitochondrial motility in vivo. (A, B)** Stills from movies of GFP-labelled mitochondria in wing neuronal axons expressing *UAS-wt-Miro* (A) or *UAS-Split-Miro* (B) in *miro^Sd32/+* background during the first minute (top panels) and fifth minute (bottom panels) of blue light exposure. Traces of transported mitochondria in corresponding movies are overlayed onto the images. **(C, D)** Number of motile mitochondria captured in a 50-μm axonal tract in wing neurons expressing *UAS-wt-Miro* (C) or *UAS-Split-Miro* (D). Bar charts show the average mitochondrial content at each time point. Filled circles represent the number of mitochondria within each axonal bundle at minute 1 and 5. Data were analysed by paired Student's *t* test. Number of wings analysed: *UAS-wt-Miro* = 8, *UAS-Split-Miro* = 8, from 2 independent experiments. **(E, F)** Anterograde velocity (E) and run length (F) of axonal mitochondria in wing neurons expressing *UAS-wt-Miro* or *UAS-Split-Miro*, during the first and fifth minute of blue light exposure, relative to (C, D). Due to the overall lower number of bidirectional and retrograde-moving mitochondria, a meaningful statistical analysis of their velocity and run length is not possible. Circles represent tracked mitochondria. Data mean ± SEM, Mann–Whitney test. *** $p < 0.001$. The data underlying the graphs shown in the figures can be found in S1 Data.

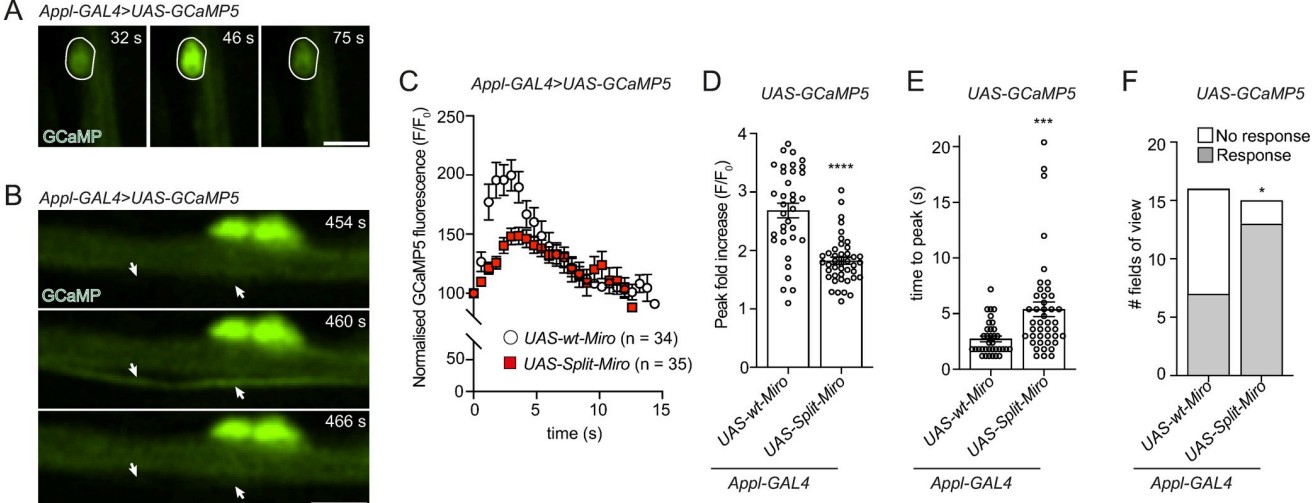

**Fig 6. Basal neuronal activity is affected by Split-Miro. (A, B)** Examples of spontaneous bouts of neuronal activity, indicated by transient fluorescence increase of the GCaMP5 $Ca^{2+}$ indicator in the cell body (A) and axonal bundle (B) of adult wing neurons. White circle, neuronal soma (A). In (B), the arrows indicate a single axon within the axonal bundle. Scale bars: 5 μm. **(C)** Pan-neuronal expression of *UAS-Split-Miro* significantly reduces the amplitude of $Ca^{2+}$ transients, compared to *UAS-wt-Miro*. Traces indicate the average GCaMP5 fluorescence intensity at individual time points in neuronal axons. $N$ = number of bouts of activity from 16 wings (*UAS-wt-Miro* = 8; *UAS-Split-Miro* = 8) and 2 independent experiments. **(D, E)** Peak of GCaMP fluorescence expressed as fluorescence fold increase from baseline (D) and time to reach the peak (E). Circles, combined bouts of activity from neuronal soma and axons, relative to (A-C). Data are mean ± SEM, Mann Whitney test. **(F)** Quantification of the number of active neuronal regions after 5 minutes of time-lapse imaging with 488-nm blue light in *UAS-wt-Miro* and *UAS-Split-Miro* flies. The number of neuronal regions (cell bodies and axons) that respond were counted in 16 (*UAS-wt-Miro*) and 15 (*UAS-Split-Miro*) fields of view, Fisher's exact test. * $p < 0.05$, *** $p < 0.001$, **** $p < 0.0001$. Refer to S3 Table for full genotypes. The data underlying the graphs shown in the figures can be found in S1 Data.

spontaneous $Ca^{2+}$ transient with the GCaMP5 $Ca^{2+}$ indicator (Fig 6A and 6B). As done for the trafficking experiment, we expressed Split-Miro and the wt-Miro control in a $miro^{+/-}$ background and recorded basal neuronal responses under exposure to blue laser light. We found that $UAS\text{-}Split\text{-}Miro^+$ neurons displayed a milder and slower response compared to $UAS\text{-}wt\text{-}Miro^+$ control neurons (Fig 6C–6E). Interestingly, the dampened neuronal firing was offset by an enlarged area of active neurons (Fig 6F), suggesting that Split-Miro photocleavage induced distributed network performance, which characterises neural ensembles and manifolds [41,42].

## Inducible hyperactivity of Split-Miro flies is exacerbated by age

Genetic mutations and RNAi have shown that Miro is critical for mitochondrial functionality in the nervous system [5,7,8]. Homozygous *miro* gene loss of function alleles are lethal, thus precluding a comprehensive analysis of Miro function in adult animals. Conditional loss of Miro1 in mouse neurons causes severe movement defects within 30 days postnatal [5], and both increased and reduced Miro abundance in the *Drosophila* nervous system can rescue fly climbing activity in models of neurodegeneration [43,44]. These findings suggest that real-time Miro disruption in the adult nervous system could be exploited to manipulate animal behaviour.

Having demonstrated that Split-Miro affects mitochondrial functionality in S2R+ cells and neuronal physiology in vivo, we turned to behavioural genetics to assess Split-Miro versatility for the study of organismal phenotypes. Pan-neuronal expression of *UAS-mCherry-MiroN-LOV2* and *UAS-Zdk1-MiroC* ("UAS-Split-Miro") with the *Appl-Gal4* driver rescued the lethality associated with classic null mutations of the Miro gene ($miro^{Sd32/B682}$) (Figs 7A and S5A).

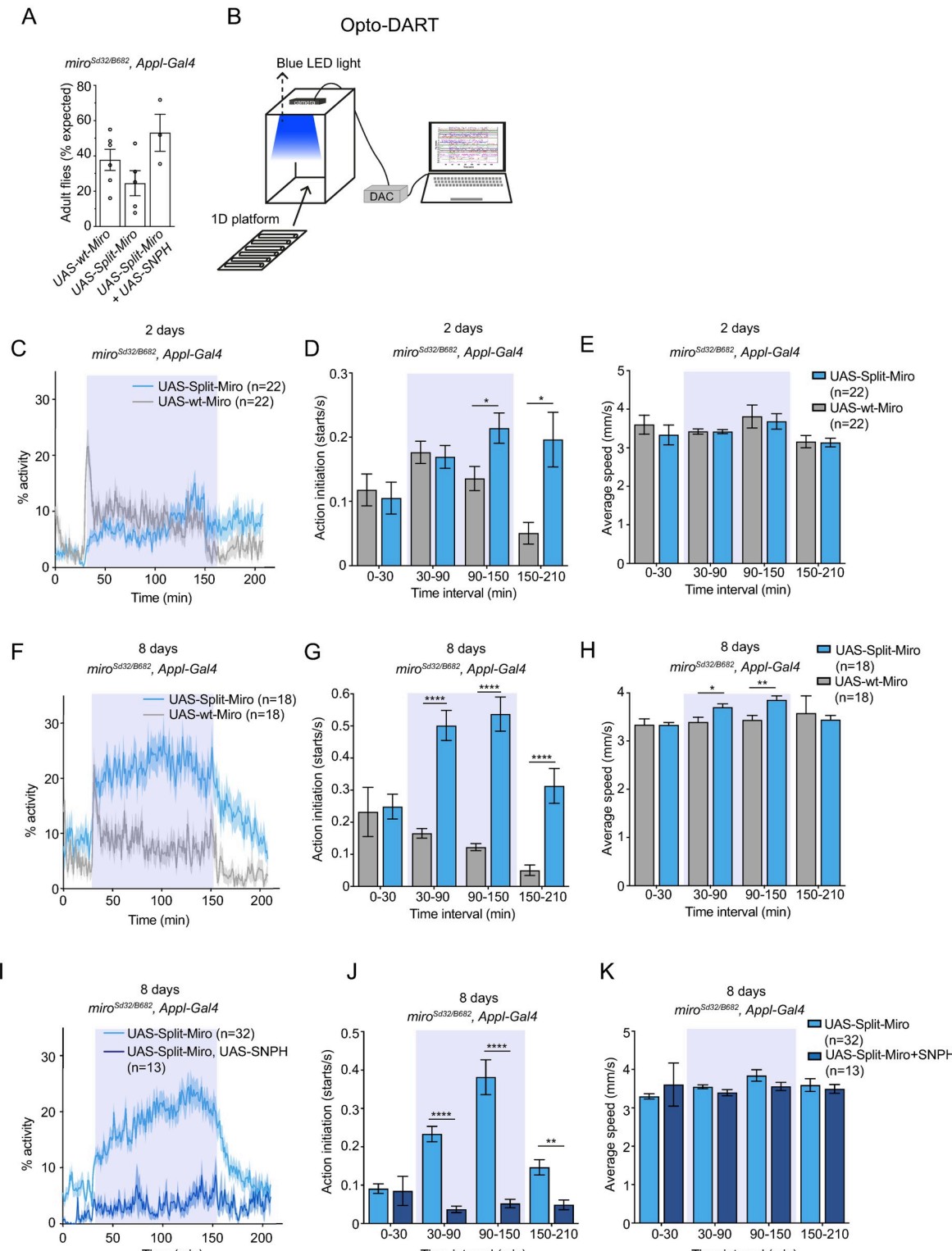

**Fig 7. Optogenetic control of *Drosophila* locomotor behaviour in Split-Miro flies. (A)** Adult *miro*$^{Sd32/B682}$ flies eclosed after *Appl-Gal4*-driven expression of *UAS-wt-Miro*, *UAS-Split-Miro* or *UAS-Split-Miro + UAS-EGFP-SNPH* (*UAS-SNPH*) were counted from 6, 5, and 3 independent crosses, respectively. Data are reported as percentage of flies expected from mendelian ratios. Data are mean ± SEM, one-way ANOVA with Tukey's post hoc test showed no statistical difference. **(B)** Schematic of the "Opto-DART" behavioural setup consisting of a custom-made optogenetic enclosure equipped with LEDs for blue light stimulation and a camera to record fly activity. Flies are transferred

to a 1D platform for automated recording of activity using the DART system [45,46]. DAC: Digital-to-Analog Converter for multiplatform integration. **(C-H)** Overall activity, action initiation, and average speed of 2-day-old flies (C-E) and 8-day-old flies (F-H) expressing UAS-wt-Miro or UAS-Split-Miro, before, during, and after blue light exposure (shaded blue rectangles). In (C) and (F), the average activity of flies expressing UAS-wt-Miro vs. UAS-Split-Miro is: before light exposure, $3.78\% \pm 0.94$ vs. $2.34\% \pm 0.72$ ($p = 0.11$) (C) and $5.62\% \pm 1.02$ vs. $8.7\% \pm 1.81$ ($p = 0.14$) (F); under blue light, $9.52\% \pm 0.96$ vs. $7.32\% \pm 0.81$ ($p = 0.13$) (C) and $8.45\% \pm 0.82$ vs. $21.87\% \pm 2.07$**** (F); after blue light exposure, $3.36\% \pm 1.18$ vs. $7.85\% \pm 1.96$ ($p = 0.13$) (C) and $3.23\% \pm 1.1$ vs. $13.19\% \pm 2.39$** (F). **(I, K)** Overall activity, action initiation, and average speed of 8-day-old flies expressing *UAS-Split-Miro* or *UAS-Split-Miro + UAS-SNPH*, before, during, and after blue light exposure (shaded blue rectangles). In (I), the overall activity of flies expressing *UAS-Split-Miro* vs. *UAS-Split-Miro + UAS-SNPH* is: $6.09\% \pm 0.89$ vs. $0.95\% \pm 0.35$**** before light exposure; $18.62\% \pm 1.4$ vs. $3.55\% \pm 0.75$**** under blue light; $9.9\% \pm 1.4$ vs. $3.91\% \pm 1.25$* after blue light exposure. In (C-K), $n$ = number of flies. Values are means ± SEM, Mann–Whitney test (C, F, I) and multiple unpaired $t$ test (D, E, G, H, J, K) with Holm–Sidak correction for multiple comparisons. * $p < 0.05$, ** $p < 0.01$, **** $p < 0.0001$. The data underlying the graphs shown in the figures can be found in S1 Data.

Adult flies were assayed for their motor behaviour at day 2 and day 8 before, during, and after blue light exposure using the "Opto-DART" system (Fig 7B). *miro*[Sd32/B682], *UAS-Split-Miro* flies at 2 days of age showed a modest increase in locomotor activity after an hour of exposure to blue light, which was sustained also after the exposure to light ceased (Fig 7C–7E). These observations suggest that the effects of Split-Mito photocleavage were not rapidly reversed after removal of blue light.

Strikingly, 8-day-old *miro*[Sd32/B682], *UAS-Split-Miro* flies displayed a rapid and sustained increase in motor activity when exposed to blue light compared to control, while no difference was observed in their baseline activity prior to exposure to blue light (Fig 7F–7H). While the fly speed returned to control level as the blue light was turned off, the action initiation revealed a prolonged effect on locomotor activity, similar to what observed in younger flies (Fig 7F–7H). Remarkably, Split-Miro–dependent effects on fly activity and action initiation could be reversed by coexpressing human SNPH (*UAS-SNPH*) with *Appl-Gal4* (Fig 7I–7K). Thus, SNPH can also suppress the effect of Miro inactivation in the context of animal behaviour. Together, these results show that acute loss of Miro function leads to fly hyperactivity, and this phenotype is exacerbated by ageing and suppressed by expression of SNPH. Collectively, these experiments reveal that Split-Miro is effective in an adult animal and uncover a previously unknown role of Miro in the regulation of *Drosophila* locomotor behaviour.

## Discussion

Using optogenetics to implement a real-time LoF paradigm by targeting Miro, we show that collapse of the mitochondrial network is an immediate response to Miro photocleavage in S2R+ cells, which temporally precedes the defects observed in mitochondrial trafficking. We found that Miro overexpression increases the proportion and the processivity of mitochondria transported in the processes of S2R+ cells. Surprisingly, although sustained Split-Miro photocleavage reverted the proportion of transported mitochondria to control levels, the velocities and run lengths of the motile organelles were largely unaffected. Interestingly, Split-Miro photocleavage decreases the number, but not the velocities and run length, of motile mitochondria in adult fly neurons in vivo, suggesting that a potentially similar mechanism might account for the regulation of mitochondrial motility in the processes of S2R+ cells and in adult neurons. This scenario is consistent with an essential role for Miro in the recruitment of transport complexes for activation of bidirectional transport, likely by recruiting [47,48] or directly activating [49] molecular motor complexes. However, we hypothesise that once motors have been recruited onto mitochondria, they may link to the organelles via additional factors in a Miro-independent way, at least on a proportion of mitochondria (Fig 2I). Future studies should focus on discovering these factors, for example, by testing if the functional homologs of metaxins [50] in *Drosophila* could fulfil this role.

We previously showed that mitochondrial motility declines with age in the axons of *Drosophila* neurons [20,38], which can be partly rescued by boosting the cAMP/PKA signalling pathways and the abundance of the kinesin-1 motor protein [38]. However, the mechanisms underlying transport decline are still not clear. In the current study, we observed that Split-Miro photocleavage decreases mitochondrial motility in 2-day-old flies while there was no effect in 7-day-old flies. This result is intriguing as it suggests that, while important to maintain transport in young flies, Miro is largely dispensable to sustain the less abundant transport typically observed at later stages [20], implying that Miro-independent mechanisms may become predominant.

Rapid retraction of the mitochondrial network in the cell soma after Split-Miro photocleavage is conceivably a consequence of releasing membrane tension that accumulates under stretch, reminiscent of the recoil of daughter mitochondria after fission [51]. Increased mitochondrial tension following Miro overexpression is consistent with the idea that more motors are recruited and pull onto the mitochondrial membrane via cytoskeletal interaction [52–55], which would likely contribute to the buildup of mechanical energy onto an interconnected network. According to this view, releasing the link between mitochondria and the microtubules then triggers the rapid collapse of the network. It would be interesting in future investigations to establish whether the phenotype that we observe could be regarded as a "mitoquake," i.e., rapid mitochondrial network disruption with associated release of mechanical energy, similar to the sudden cytoskeletal rearrangements ("cytoquakes") that were proposed to underpin mechanical adaptivity during cellular dynamic processes [56].

We found that using SNPH to tether mitochondria onto the microtubule network prevented Split-Miro–induced mitochondrial reorganisation, indicating that Miro stabilises the mitochondrial network by providing an anchor to the cytoskeleton. It is possible, however, that Miro may stabilise the mitochondrial network by simultaneously bridging mitochondria to different cellular structures. In this regard, we did not find that the actin cytoskeleton plays a significant role in S2R+ cells (S6 Fig), although it is conceivable that the actin network contributes to mitochondrial stability via Miro-Myosin interactions in other cell types [6,57–59]. We showed that Split-Miro photocleavage does not affect mitochondria-ER contacts using a split-GFP reporter, suggesting that the associations mitochondria establish with the ER do not significantly contribute to the stability of the mitochondrial network. Testing whether Miro-dependent mitochondrial interaction with other cellular structures is necessary to maintain mitochondrial network stability is a goal for future studies.

Decreasing the abundance of Miro by RNAi reduced $[Ca^{2+}]_m$ levels in the neurons of *Drosophila* brain [11,12], and mutating the $Ca^{2+}$-binding EF domains of Miro reduced $[Ca^{2+}]_m$ uptake in mouse hippocampal neurons [60], although Miro1-KO and Miro-EF mutant MEFs did not show any disturbances in $[Ca^{2+}]_m$ homoeostasis [5,27]. In S2R+ cells, shedding Split-Miro functional domains (including the $Ca^{2+}$-binding EF-hands motifs) did not have any effect on $[Ca^{2+}]_m$ uptake. We hypothesise that impaired $[Ca^{2+}]_m$ homeostasis shown with classical Miro LoF approaches (i.e., RNAi, knockout) may be a secondary effect of Miro LoF, potentially a consequence of sustained morphological and transport defects of mitochondria. However, because Miro was shown to interact with MCU and the Sam/MICOS complexes [31,32], presumably via direct interaction with the Miro transmembrane domain, we cannot exclude that the short mitochondrial targeting sequence might still mediate $[Ca^{2+}]_m$ homeostasis. Our finding that MERCS were not affected by Split-Miro photocleavage supports the idea that $[Ca^{2+}]_m$ uptake, known to be regulated by MERCS, is not a primary role of Miro in this context.

Overexpressing SNPH to rescue Split-Miro–dependent mitochondrial network retraction also did not affect $[Ca^{2+}]_m$ uptake. Because SNPH also locks mitochondria into a stationary

state with little network dynamic, these results raise the intriguing possibility that mitochondrial movements are not critical to maintain $[Ca^{2+}]_m$ homeostasis, as long as mitochondria maintain their functionality.

Elegant methods for light-induced repositioning of trafficked vesicles and mitochondria have been developed, which are based on the recruitment of truncated forms of motor proteins to overpower the endogenous transport machinery and so to achieve controlled redistribution of cellular cargoes [61–64]. Engineering the LOV2-Zdk1 domain into subunits of motor and adaptor proteins could offer a complementary strategy for studying intracellular trafficking when a real-time LoF approach is preferred. Because protein photocleavage is reversible, the LOV2-Zdk1 methodology also offers significant advantages over existing methods based on the rapid, nonreversible, degradation of a target protein by the proteasome, such as the degron [65] or the TRIM-away systems [66].

Opsin-based optogenetic approaches to activate/repress specific neurons and study associated animal behaviour have been extensively used in live animals [67]. A LOV2-controlled CaMKII inhibitor was used to impair memory formation in live mice after blue light stimulation for 1 hour [68]. By creating Split-Miro flies, we combine optogenetics with *Drosophila* behaviour and neuronal specificity to perform LoF experiments in adult animals in real time. Although the locomotor behaviour of Split-Miro flies is indistinguishable from the wild-type counterpart before exposure to blue light, their activity is enhanced under blue light, and this hyperactivity becomes more pronounced with age.

The mechanisms underlying the exacerbated hyperactivity phenotype in older flies are unexplained. It is conceivable that the alteration of neuronal activity observed in young Split-Miro flies is linked to an acute imbalance of synaptic transmission and leads to augmented locomotor activity. We speculate that, in older flies, a potentially compromised cellular state might contribute to amplify this response. In line with this notion, ageing has been associated with increased activity of excitatory neurons in *C. elegans*, flies, and mice [69–72]. We are mindful that we performed the neuronal activity experiments in the wing neurons of the flies, and, although these cells can relay signals to affect motor phenotypes [73,74], we are cautious not to generalise our findings to all neurons in the fly.

The observed reduction in mitochondrial axonal transport induced by Split-Miro may also be a contributing factor towards enhanced fly activity, at least in young flies, by causing an imbalance in neuronal activity. It is known that reducing mitochondrial number positively correlates with activity-dependent vesicular release at the presynapses of hippocampal and cortical neurons [10,39] and with miniature excitatory junction potentials at the *Drosophila* NMJ [4]. In this view, the remarkable rescue of Split-Miro–induced hyperactivity in older animals by SNPH overexpression could conceivably occur via further reduction of the remaining mitochondrial transport in older neurons or via a direct effect on synaptic mitochondria [40]. Overall, our data point to an important role of neuronal mitochondrial mobility for animal behaviour and suggest that Miro could play a crucial role in preventing hyperexcitation in the ageing nervous system with potential ramification in the context of neurodegeneration.

## Materials and methods

### Generation of plasmid DNA

The new constructs produced in this study are reported in S1 Table and were generated either through restriction enzymes mediated cloning or NEBuilder HiFi DNA Assembly (NEB) using the primers listed in S2 Table. The plasmid inserts were amplified by PCR using the Q5 Hot-Start High-Fidelity 2X Master Mix (NEB). Site-directed mutagenesis was performed using the Q5 Site-directed mutagenesis kit (NEB) following the manufacturer's instructions.

The fidelity of all assembled constructs was verified by Sanger sequencing (Eurofins Genomics).

## Isolation of *Drosophila* Miro cDNA

Five Oregon-R flies were anesthetised on dry ice, rapidly grinded to powder with a plastic pestle in an Eppendorf tube and the total RNA extracted with the RNeasy Micro kit (Qiagen) following the manufacturer's instructions. Reverse transcription was performed with the iScript Select cDNA Synthesis kit (BioRad), using 1 μl of total RNA and Oligo dT primers. *Drosophila* Miro is encoded by a single gene, located on the third chromosome, from which potentially 4 different transcripts are produced (FlyBase reference: FBgn0039140). The coding sequence of the longest Miro-RE/Miro-RF isoform was amplified from the total *Drosophila* cDNA with the primer pair #1 and #2 (S2 Table) and was used as a template to engineer wt-Miro and the Split-Miro variants employed in this study.

## Cell culture, transfection, and RNAi

*Drosophila* S2R+ cells were obtained from the *Drosophila* Genomics Research Centre (DGRC, Indiana University) and cultured in Schneider's insect medium (Gibco) at 25˚C. Cells were transfected using FuGene HD transfection reagent (Promega) using a DNA:FuGene ratio of 1:3 following the manufacturer's instructions.

For RNAi experiments, 532 bp (Miro) and 599 bp (Control) dsRNA molecules were transcribed with the MEGAScript RNAi kit (Thermo Scientific) from templates generated by PCR of sequences within the Miro 3′-UTR (targeting only endogenous, and not overexpressed, Miro and Split-Miro) and the pT2-DsRed-UAS plasmid backbone, respectively. Primers used are listed in S2 Table. Cells were treated with 15 μg/ml dsRNA for a total of 6 days, replacing the dsRNA every 24 hours in fresh media, prior to imaging.

## *Drosophila* husbandry and generation of transgenic flies

The fly strains used in this study are listed in S3 Table. Flies were maintained on "Iberian" food [70 mg/ml yeast (Brewer's yeast, MP Biomedicals, 903312), 55 mg/ml glucose (VWR, 10117HV), 7.7 mg/ml agar (SLS, FLY1020), 35 mg/ml organic plain white flour (Doves Farm, UK), 1.2 mg/ml Nipagin (Sigma, H3647), 0.4% propionic acid (Sigma-Aldrich, P5561] at 25˚C and 60% humidity with a 12-hour light–12-hour dark cycle.

The transgenic fly lines generated in this study were obtained by phiC31-mediated transgenesis to integrate the relevant constructs into either the attP40 (25C6) or attP2 (68A4) landing sites following embryo injection.

## Immunoprecipitation

Cells were transfected in 10-cm dishes, washed with PBS, and incubated with 900 μl IP lysis buffer (50 mM Trizma (pH 7.4), 150 mM NaCl, 5 mM EDTA, 0.5% Triton X-100, 1× PhosSTOP, 1× cOmplete EDTA-free protease inhibitor cocktail) on ice for 20 minutes. The cell lysate was homogenised through a 23G syringe needle and centrifuged at 21,000*g* for 30 minutes at 4˚C. The cleared lysate (1 mg) was incubated at 4˚C overnight with 25 μl of GFP-Trap beads (Chromotek) pre-equilibrated by washing 3x times in IP lysis buffer. After incubation, the beads were washed 3x times with IP buffer, and the immunoprecipitated material was eluted in 20 μl of 2X NuPAGE sample buffer supplemented with 40 mM DTT for 10 minutes at 95˚C.

## Western blotting

Samples were loaded on a NuPAGE 4% to 12% Bis-Tris protein gel and transferred onto an Immobilion PVDF membrane (Merk-Millipore) for 1 hour at 35 V in a wet transfer apparatus buffered with 12.5 mM Trizma-Base (Sigma), 96 mM glycine (Sigma), 10% methanol (Fisher Chemicals). The membrane was incubated in blocking solution [5% milk in PBST (PBS (pH 7.4) with 0.1% Tween-20 (VWR))] for 2 hours and probed overnight with the following primary antibodies: anti-β-tubulin (1:100, Developmental Studies Hybridoma Bank, DSHB #7, AB_2315513), anti-dMiro [75] (1:50,000), anti-Milton [76] (1:1,000, monoclonal antibody 2A108), anti-GFP (1:1,000, Chromotek, PABG1). Membranes were incubated for 1 hour with either an IRDye secondary antibody (for LI-COR Odyssey imaging) or with horseradish peroxidase (HRP)-conjugated secondary antibodies followed by a 3-minute incubation with a chemiluminescent substrate (GE Healthcare) for ChemiDoc imaging. Secondary HRP-conjugated antibodies: anti-rabbit (1:5,000, NIF824), anti-mouse (1:5,000, NIF825), anti-guinea pig (1:5,000, SeraCare 5220–0366). LI-COR secondary antibody: IR Dye 800CW goat anti-mouse IgG (1:10,000).

## Live cell imaging of cargo transport and photostimulation

S2R+ cells were seeded in 4-well Nunc Lab-Tek chambered coverglass (Thermo Scientific) coated with 0.5 mg/ml Concanavalin A (Sigma). To induce the formation of cellular processes, cells were treated with 1 μM cytochalasin D (Sigma) for 3 to 4 hours before imaging and throughout the imaging period. Before imaging, cells were stained with the following dyes, depending on the specific experimental setting: 200 nM MitoTracker Green FM, 200 nM MitoTracker DeepRed FM (Thermo Scientific), 100 nM MitoView 405 (Biotium), 1X TubulinTracker DeepRed (Thermo Scientific), 1X ActinTracker (CellMask Actin Tracker Stain, Invitrogen). The MitoView 405 dye was added to the cells 15 minutes prior imaging and kept throughout. All the other dyes were washed off prior to imaging, after 30 minutes of incubation. Live cell imaging was performed using a Nikon A1RHD inverted confocal microscope, equipped with a photostimulation unit, GaAsP and multialkali PMT detectors, 405-nm, 488-nm, 561-nm, and 640-nm laser lines and a Nikon 60X/1.4NA Plan Apochromatic oil immersion objective, unless specified otherwise. Experiments were carried out at a constant 25˚C temperature in an Okolab incubation system, and time series were digitally captured with the Nikon NIS-Elements software at 1 frame per seconds (fps), unless specified otherwise.

Sustained Split-Miro photocleavage in S2R+ cells was achieved with a 0.3% (0.017 mW) 488-nm laser, scan zoom of 4, at a framerate of 1 fps for the entire duration of the experiment. For dual colour imaging during sustained Split-Miro photocleavage, a frame every 1 minute was acquired with either the 405-nm, 561-nm, or 640-nm laser line. Transient whole-cell photocleavage of Split-Miro with untagged C-terminus was achieved with a stimulation step of 570 ms using a 4% 488-mn laser at a scan speed of 4 fps and 2 scanning iterations. In these experiments, a Nikon 40X/1.15NA Apochromatic water immersion objective with a scan zoom of 4 was used. The stimulation area was set to 492.7 μm$^2$, sufficient to cover the whole cell area. Time series before and after stimulation were acquired at 0.5 fps. To achieve transient photocleavage of double-tagged Split-Miro, cells were imaged with both a 488-nm (0.8%, scan time 0.6 seconds/frame) and a 561-mn (1%, scan time 0.6 seconds/frame) laser for 8 seconds with a frame rate of 0.5 fps, scan zoom of 4. Recovery was monitored by imaging with a 561-nm laser with a frame rate of 0.5 seconds for up to 3 minutes.

To combine optogenetic stimulation with 3-colour imaging and fast acquisition (as in Fig 3), the same settings were maintained, and the 640-nm laser power was set to 0.8%.

Imaging the neurons of the adult *Drosophila* wing was performed as reported in Vagnoni and colleagues [37,38]. For rapid blue light photostimulation in this tissue, a 5% (0.184 mW) 488-nm laser, scan zoom of 4 was used with a scan speed of 4 fps and 4 scanning iterations. The total duration of the stimulation step was 1.14 seconds, and the stimulation area was set to cover a 46.23 $\mu m^2$ region of the neuronal axons.

Mitochondrial trafficking in adult wing neurons was recorded with a frame rate of 0.5 frames/seconds for 5 minutes and quantified from the bundle of marginal axons along the L1 vein, essentially as we described previously [37,38]. The in vivo trafficking experiments were performed using a Nikon spinning disk system with a CSU-X1 scanning head (Yokogawa) and an inverted microscope stand (Eclipse Ti-E (Nikon)) equipped with a 60 × CFI Apo oil immersion objective (1.4 NA) and an EM-CCD camera (Du 897 iXon Ultra (Andor)). Imaging of mitochondrial motility and concomitant Split-Miro photocleavage were achieved using a 15% 488-nm laser light and 400-ms light exposure.

### Analysis of mitochondrial and peroxisomal transport in S2R+ cells

Organelle motility was quantified in 2 to 6 processes for each cell. Only processes that remained in the same focal plane throughout acquisition and had no or minimal crossing with other processes were selected. Each process was straightened using the "Straighten" function in Fiji/ImageJ and kymographs produced and analysed with the "Velocity Measurement Tool" (https://dev.mri.cnrs.fr/projects/imagej-macros/wiki/Velocity_Measurement_Tool) in Fiji/ImageJ.

"Run" was defined as a continuous movement that lasts for at least 3 seconds without any change in direction or velocity. Organelles were defined as "motile" if engaging in at least one run $\geq 2$ $\mu m$. A "pause" was defined as a period of at least 3 seconds during which the mitochondrion does not change its position. The duty cycle describes the motile behaviour of the organelles as percentage of time spent on long runs ($\geq 2$ $\mu m$), short runs ($< 2$ $\mu m$) and pausing, or in the anterograde and retrograde direction.

### Analysis of mitochondrial morphology and mitochondria-ER contacts

Mitochondrial morphology in the cell soma was analysed from cells showing a clearly interconnected network before photocleavage, using Fiji/ImageJ and a pipeline adapted from Chaudhry and colleagues [77]. Briefly, the "Adaptive Threshold" command was used to create binary images that were individually checked for accuracy in marking the mitochondrial network. The "Analyse particles" command was used to measure mitochondrial size, with a cutoff set at 0.09 $\mu m^2$. A skeleton of the mitochondrial network within each cell was obtained and analysed with the "Skeletonize" and "Analyze skeleton (2D/3D)" functions, respectively. The same analysis pipeline was used to quantifying mitochondria-ER contacts by creating a mask of the SPLICS signal and counting the GFP puncta throughout the cell.

Mitochondrial mass was defined as the total area covered by mitochondria within the cell and normalised to the first time point. The mitochondrial aspect ratio (AR) was measured in single mitochondria smaller than 6 $\mu m^2$ and was defined as the ratio between the longest and the shortest axis of the organelle. For the analysis of the mitochondrial network branching, only branches emanating from the main mitochondrial skeleton were considered.

### Analysis of Split-Miro kinetics

The kinetics of the cytoplasmic release of mCherry-tagged Split-Miro N-terminus after photocleavage and the reconstitution at the mitochondria was analysed by measuring the cytosolic fluorescence intensity of the mCherry before and after photocleavage. For cell culture

experiments with single-tagged Split-Miro, images were analysed with the General Analysis tool of the Nikon NIS-Elements Software by defining a region of interest (ROI) covering the cellular cytosol (with no mitochondria) and measuring the fluorescence intensity at each time point. For cell culture experiments with double-tagged Split-Miro and for neuronal in vivo experiments, images were analysed with Fiji/ImageJ by measuring the fluorescence intensity of a cytosolic ROI devoid of mitochondria (area is 0.451-$\mu$m$^2$ and 0.126-$\mu$m$^2$, respectively). For each cell or fly wing, the fluorescence intensity over time was normalised to a scale from 0 to 1, where 0 corresponds to the minimum value measured before Split-Miro N-terminus release and 1 to the maximum value measured after photocleavage. Mitochondria/cytosolic ratio of both mCherry-tagged Split-MiroN and EGFP-tagged Split-MiroC in S2R+ cells were calculated by measuring the fluorescence intensity at different time points from two 0.451-$\mu$m$^2$ ROIs located within the mitochondrial network and within a proximal area of the cytosol devoid of mitochondria. An mCherry ratio of 1 indicates the signal is homogeneous throughout the cytoplasm, and, thus, mCherry-Split-MiroN has reached complete release.

## Live cell calcium imaging in S2R+ cells

S2R+ cells were imaged with a Nikon Ti-E inverted epifluorescence microscope equipped with a Nikon 60X/1.4NA Plan Apochromatic oil immersion objective, a mercury lamp (Nikon Intensilight C-HGFI) for illumination, and a Dual Andor Neo sCMOS camera for detection. Cells were plated on a Concanavalin A-coated 18-mm coverslip and imaged in a Ludin imaging chamber (Type 1, Life Imaging Services) in S2 media. Cells were imaged at 1 fps for 3 minutes under blue light, which was sufficient to induce Split-Miro–dependent mitochondrial network retraction, prior to ionomycin stimulation. A peristaltic pump system (Ismatec) was used to perfuse the imaging chamber with 2.5 $\mu$M ionomycin (Thermo Fisher Scientific) until all the media in the chamber was replaced. Additional 5-minute imaging at 1 fps was performed to record the dynamic mito-GCaMP6 signal. To verify Split-Miro cleavage, a frame with green light (to capture mCherry-Split-MiroN) was acquired at the beginning and the end of each acquisition period.

To quantify mitochondrial calcium, an ROI was drawn to outline single cells and the fluorescence intensity of mito-GCaMP6f signal measured using the Nikon NIS-Elements software. Response curves were aligned at the base of the response peak, and, for each cell, the fluorescence intensity at every time point was normalised to the value of the first frame. Only cells that responded at least 2-fold to ionomycin in the first 100 seconds were included in the quantification. Mito-GCaMP6f "peak" was defined as the maximum fold-change value reached during imaging. The time to reach the peak was calculated as the $\Delta t$ between the "peak" time point and the time of first response to stimulation.

## Live cell calcium imaging in adult neurons

Calcium transients in the wing marginal neurons were acquired using the same spinning disk imaging system described to capture mitochondrial trafficking. Image series were recorded for 5 minutes using a 10% 488-nm laser light with an acquisition time of 2 frames/seconds and exposure time of 200 ms. Image series from 1 to 2 fields of view were acquired from the same areas of each wing (i.e., mid-margin to the L1 to L2 vein intersection). To quantify the GCaMP fluorescence during spontaneous calcium transients, an ROI was drawn around each responding neuronal soma and axons. After background subtraction in Fiji/ImageJ (rolling ball radius = 50 px), the ROI mean intensity over time was calculated using the Plot Z-axis profile command. GCaMP fluorescence peak and time to reach the peak were calculated as done for calcium imaging in S2R+ cells.

## Behavioural assay

Flies were anesthetised with $CO_2$ on the day of eclosion, transferred to new food, and aged for either 2 or 8 days prior to the assay. On the day of the assay, flies were briefly anesthetised on ice and individually loaded into 65-mm glass tubes on custom-made platforms. Fly activity was recorded in an optogenetic enclosure (BFK Lab) equipped with LEDs for 444-nm blue light stimulation and a webcam for video recording. The position of each fly was automatically tracked and the quantification of the fly locomotor activity performed using the *Drosophila* ARousal Tracking (DART) system [45,46]. Flies were left to acclimatise for 20 minutes before starting the experiment. Fly activity was recorded continuously for 30 minutes before the blue LED light was switched on for 2 hours (power at the platform: 5 mW/cm$^2$). After the blue light was switched off, fly movements were recorded for an additional hour, for a combined experimental time of 3 hours, 30 minutes. Flies were defined as "active" if displaying bouts of activity of at least 2 mm/seconds. The active average speed refers to the speed of fly movement when active. The action initiation is defined as the number of times a bout of activity is started per second. The % activity represents the percentage of time the fly spends moving.

## Statistical analysis and image preparation

Data were analysed with Microsoft Excel and GraphPad Prism 9. Statistical tests and number of replicates are reported in the figure legends. The data underlying the graphs shown in the figures can be found in S1 Data. Images were assembled using Fiji/ImageJ. For presentation purposes, "Subtract Background" (rolling ball radius = 20 pixels), and either the "Despeckle" or "Smooth" filters were used to reduce salt and pepper noise. All images in the same experimental series were processed in the same manner.

## Supporting information

**S1 Raw Images. Panels in "Supporting Figure 2" (A) and (D) include original blots used in S2(A) and S2(D) Fig, respectively.** In the top panel in (A), the faint double bands between the 102 and 150 molecular weight markers indicate nonspecific binding of the Miro antibody. In the middle panel in (D), the bands of approximately 76 kDa correspond to endogenous Miro. The absence of endogenous Miro in the "IP" lanes indicate that transfected Split-Miro does not bind to endogenous Miro at high affinity. Panels in "Supporting Figure 5" include original blots used in S5(A) Fig. The band above the 150 molecular weight marker indicates nonspecific binding of the Miro antibody. In all panels, "X" above a lane indicate that lane is not included in the final figure.
(PDF)

**S1 Data. Excel spreadsheet containing the numerical data and details of statistical analysis for Figs 1D, 1E, 1F, 1G, 2C, 2D, 2F, 2G, 2H, 3B–3D, 3F, 3G, 4B, 4C, 4D, 4E, 4G, 4H, 5C, 5D, 5E, 5F, 6C, 6D–6F, 7A, 7C, 7D, 7E, 7F, 7G, 7H, 7I, 7J, 7K, S1C, S1D, S1F, S1G, S2B, S2C, S2G, S2H, S2I, S2J, S2K, S3A, S3C, S3D, S3F, S3G, S3I, S4B, S5C, S5D, S5E, S5F, S5G and S5H.**
(XLSX)

**S1 Fig. Split-Miro reconstitution and kinetics of Split-Miro harbouring wild-type and mutant LOV2 proteins. (A)** EGFP-Split-Miro-mCherry is reconstituted and localises at the mitochondria in the absence of sustained blue light irradiation. S2R+ cells were cotransfected with mCherry-tagged Split-MiroC (magenta) and EGFP-tagged Split-MiroN (green), mitochondria are stained with MitoTracker Deep Red (MTDR, cyan). Note the EGFP/mCherry tags are appended at different termini compared to the mCherry-Split-Miro-EGFP shown in

Fig 1. **(B, E)** Localisation of mCherry-Split-MiroN before and after a 570-ms pulse of blue light. S2R+ cells are cotransfected with untagged Split-MiroC and mCherry-Split-MiroN containing the T406A, T407A mutations in the N-terminus of LOV2 (LAAA, in B) or mCherry-Split-MiroN containing the wild-type sequence (LATT, in E). Before irradiation, Split-Miro is reconstituted at the mitochondria, indicated by the colocalisation of mCherry-Split-MiroN (grey) and the MitoTracker Deep Red (MTDR) staining. Immediately after irradiation (18 seconds), Split-MiroN is released into the cytoplasm (indicated by a more homogenous grey colour), and it fully reconstitutes within 2 minutes. **(C, D)** Quantification of mCherry-Split-MiroN half-time release (C) and recovery (D) after photocleavage, relative to (B). In (D), left panel shows levels of cytosolic Split-Miro N-terminus quantified after the maximum release is reached; right panel: recovery half-time. Solid line in (D) is exponential curve fit. In **(B)** and **(E)**, the cartoon depicts photocleavage and reconstitution of mCherry-tagged Split-Miro. s, seconds. Scale bar: 10 μm. **(F, G)** Comparison of mCherry-Split-MiroN release (F) and recovery (G) half-time after photocleavage shows no significant difference between the 2 LOV2 variants (unpaired Student's *t* test). The LATT (wild-type) to LAAA mutation in the N-terminus of LOV2 has been reported to have a stabilising effect on the Jα helix in cultured cells at 37°C [17,18]. We did not find any noticeable difference in the steady-state reconstitution efficiency of the 2 variants in S2R+ cells cultured at 25°C. Data are shown as mean ± SEM. Circles, number of cells, from 2 independent experiments. Scale bar: 10 μm (A) and 5 μm (B, E). The data underlying the graphs shown in the figures can be found in S1 Data.
(TIF)

**S2 Fig. Insight into the regulation of mitochondrial motility in the processes of S2R+ cells by endogenous Miro and Split-Miro. (A)** Representative western blots of Miro from total lysates of control and Miro RNAi-treated S2R+ cells. **(B)** Duty cycle analysis describes the average time mitochondria spend moving anterogradely, retrogradely, or pausing. For each parameter, all mitochondrial values from each cell were averaged and compared between control and Miro dsRNA condition using a multiple Student's *t* tests. Number of mitochondria analysed are in brackets from 29 (Ctrl dsRNA) and 36 (Miro dsRNA) cells, respectively, from 2 independent experiments. **(C)** Run velocities of short and long runs in control dsRNA-treated S2R+ cells showing that Miro-dependent long runs are significantly more processive than the short, Miro-independent runs (Fig 2C). Number of runs analysed are in brackets, from 2 independent experiments. Mann–Whitney test. **(D)** Split-Miro interacts with Milton in S2R+ cells. Cells were transfected with Split-Miro or Control (as shown in E) and the total cell lysates immunoprecipitated using GFP-beads to pull down EGFP-tagged Split-Miro C-terminus. Immunoprecipitates were blotted and probed with anti-GFP antibody (to detect Split-Miro C-terminus), anti-Miro antibody (to detect Split-Miro N-terminus), and an anti-Milton antibody. Inputs are total lysates (25 μg protein). **(E)** Cartoon showing Split-Miro and Control constructs with the GFP and Miro antibodies used for immunoprecipitation and western blotting in (D). **(F)** Representative kymographs of mitochondrial transport in the processes of S2R+ cells transfected with mCherry-tagged Zdk1-MiroC (Control), mCherry-Miro (wt-Miro), and mCherry-Split-Miro (Split-Miro). Scale bars: 2 μm (distance) and 5 seconds (time). **G)** Distribution of mitochondria run lengths in the processes of S2R+ cells, transfected with control, wt-Miro, and Split-Miro, as shown in F. *N* = number of mitochondrial runs. One-way ANOVA with Tukey's post hoc test. **(H)** Duty cycle analysis describing the average time mitochondria spend moving anterogradely, retrogradely, or pausing in control, wt-Miro, and Split-Miro–transfected cells, relative to (F). For each parameter, all mitochondrial values per cell were averaged and compared by one-way ANOVA followed by Tukey's post hoc test. Number of mitochondria are in brackets from 16 (control), 15 (wt-Miro), and 15 (Split-Miro) cells,

respectively, from 3 independent experiments. **(I)** S2R+ cells transfected with wt-Miro and Split-Miro were imaged for 1 minute with a 561-nm laser, to capture the mCherry signal, followed by 1-minute imaging with 488-nm blue light, to capture the EGFP signal after Split-Miro photocleavage (relative to Fig 2E). Number of mitochondria are in brackets from 11 (wt-Miro) and 17 (Split-Miro) cells, from 3 independent experiments. Data are shown as mean ± SEM. Kolmogorov–Smirnov test showed no statistical difference between groups. **(J)** Distribution of mitochondrial run lengths in the anterograde direction in the processes of S2R+ cells, during the first and seventh minute of time-lapse imaging with blue light in cells transfected with wt-Miro or Split-Miro. $N$ = number of runs. Mann–Whitney test showed no statistical difference between groups. **(K)** Run velocities for long processive anterograde and retrograde runs in S2R+ cells transfected with Split-Miro and Miro dsRNA (which targets endogenous Miro) and imaged by time-lapse for 7 minutes under blue light. Circles, number of runs, from 2 independent experiments. Mann–Whitney test shows no difference between first and seventh minutes of imaging, with the velocities remaining high compared to non-transfected condition (e.g., Figs 2H and S2C). This result shows that the velocities of the processive mitochondria, augmented as a consequence of Split-Miro overexpression, remain elevated even after reduction of endogenous Miro, suggesting Miro is not necessary for maintaining mitochondrial velocities. * $p < 0.05$, ** $p < 0.01$, **** $p < 0.0001$. The data underlying the graphs shown in the figures can be found in S1 Data.
(TIF)

**S3 Fig. Extended characterisation of the effects of Split-Miro photocleavage and SNPH overexpression in S2R+ cells. (A)** Quantification of the total area covered by the mitochondria within the cell. Each measurement was normalised to the average group value (wt-Miro, Split-Miro) at time point 0. Comparison across time points was performed by repeated measures one-way ANOVA followed by Tukey's post hoc test. Data are reported as mean ± SEM. Circle, number of cells, from 3 independent experiments. **(B)** S2 cells are transfected with EGFP-tagged Split-Miro N-terminus (green) and stained with MitoTracker DeepRed (MTDR, cyan). Middle panel: white and magenta stars indicate untransfected and transfected cells, respectively. Scale bar: 10 μm. **(C)** Duty cycle analysis describes the average time peroxisomes spend on long runs, short runs, or pausing. For each parameter, all peroxisomal values from each cell were averaged and compared between time points. Statistical significance was evaluated by multiple Mann–Whitney tests. **(D)** Bar chart shows the average peroxisomal content at minute 1 and 7 of time-lapse imaging with blue light. Circles represent the number of peroxisomes within each process. Statistical significance was evaluated by Wilcoxon test. In (C, D), number of processes and cells: wt-Miro = 24, 11, Split-Miro = 39, 16, from 3 independent experiments. There is no significant difference in the motility and number of peroxisomes in each process between timepoints. **(E)** Representative images showing S2R+ cells transfected with Split-Miro and imaged by time-lapse with blue light for 7 minutes. Exposure to blue light (to induce Split-Miro photocleavage) leads to altered mitochondrial morphology and distribution, without noticeable disorganisation of the microtubule network as detected by the Tubulin Tracker (magenta). Scale bar: 10 μm. Not shown, mCherry-tagged Split-Miro N-terminus. **(F)** Bar chart shows the average mitochondrial content at minute 1, 3, and 7 of time-lapse imaging with blue light in the processes of S2R+ transfected with either wt-Miro or Split-Miro. Circles represent the number of mitochondria within each process at minute 1 and 7. Data were analysed by Friedman test with Dunn's multiple comparison test. Number of processes: wt-Miro = 39, Split-Miro = 50 from 11 and 17 cells, respectively, from 3 independent experiments. **(G)** Length of cellular processes imaged for 7 minutes under time-lapse exposure to 488-nm blue light to achieve Split-Miro photocleavage. Circles, number of the processes.

Number of cells: wt-Miro = 11, Split-Miro = 17, from 3 independent experiments. Statistical significance was evaluated by Mann–Whitney test. **(H)** Representative kymographs from processes of cells transfected with an empty vector (Control) or EGFP-SNPH (SNPH). Mitochondria were stained with MitoTracker Deep Red (MitoTracker). Scale bars: 2 μm (distance) and 10 seconds (time). **(I)** Percentage of motile mitochondria in cellular processes of control or SNPH-expressing cells. Number of processes: control = 9, SNPH = 9, from 8 cells, from 2 independent experiments. Data are shown as mean ± SEM and were analysed by unpaired Student's *t* test. \**p* < 0.05, \*\*\*\* *p* < 0.0001. The data underlying the graphs shown in the figures can be found in S1 Data.
(TIF)

**S4 Fig. Split-Miro photocleavage does not affect mitochondrial membrane potential in S2R+ cells. (A)** Representative images of cells cotransfected either with wild-type Miro (mCherry-wt-Miro, EGFP-mito) or Split-Miro (mCherry-Split-Miro-EGFP). EGFP signal is used to mark the mitochondria over time; mCherry, not shown. MitoView 405 was used to monitor the mitochondrial membrane potential during the imaging period. Scale bar: 10 μm. **(B)** Ratio of MitoView and EGFP mitochondrial fluorescence intensity at the time points indicated. Number of cells: wt-Miro = 7, Split-Miro = 7, from 2 independent experiments. Data are shown as mean ± SEM. Each group was analysed by repeated measures one-way ANOVA followed by Dunnett's post hoc test, with each time point compared to the t = 0 minutes. The data underlying the graphs shown in the figures can be found in S1 Data.
(TIF)

**S5 Fig. Extended characterisation of Split-Miro in vivo. (A)** Western blots of lysates from male fly heads of the reported genotypes confirms the expression of *UAS-wt-Miro* and *UAS-Split-Miro*, in either Miro heterozygous (*miro^{Sd32/+}*) and null backgrounds (*miro^{Sd32/B682}*), using the Appl-Gal4 driver. *UAS-wt-Miro* is expressed from attP2; the *UAS-MiroN* and *UAS-MiroC* (to reconstitute Split-Miro) were expressed from attP40 and attP2, respectively. The higher molecular weight of wt-Miro and Split-Miro compared to endogenous Miro is consistent with the presence of the mCherry tag (wt-Miro) and mCherry/LOV2 tags (Split-Miro). Please note that the lower expression of Split-Miro compared to wt-Miro does not significantly affect the proportion of rescued flies, as shown in Fig 7A. The top membrane was blotted with an anti-Miro antibody recognising an N-terminal epitope (see also S2E Fig). **(B)** Representative images of axons in the L3 vein of the adult live fly wing. For colocalisation experiments, flies express mCherry-tagged wt-Miro (attP40) and EGFP-tagged Miro C-terminus (attP2). Split-Miro flies express the N-terminus and C-terminus halves of Split-Miro from attP40 and attP2, respectively. All constructs were expressed under the control of the Appl-Gal4 driver in a *miro^{+/+}* background. Arrows highlight examples of colocalised signal. Scale bar: 10 μm. **(C)** Quantification of the kinetics of Split-Miro recovery after photocleavage in the neurons of the adult fly wing in vivo. UAS-mCherry-Split-MiroN and UAS-MiroC were both expressed from attP40, using the Appl-Gal4 driver. The first time point corresponds to maximum levels of mCherry-Split-MiroN release. Data are shown as mean ± SEM. Red solid line, exponential curve fit (*n* = 6 wings, from 3 flies). **(D)** There is no difference in the half-life recovery of Split-Miro after photobleaching in S2R+ cells (in vitro) and in the neurons of the adult fly wing (in vivo). Data are shown as mean ± SEM and were analysed by Mann–Whitney test. Circles, number of cells (in vitro) and wings (in vivo) analysed. **(E, F)** Number of motile mitochondria captured in a 50-μm axonal tract in 7-day-old wing neurons expressing wt-Miro (E) or Split-Miro (F). Bar charts show the average mitochondrial content at each time point. Filled circles represent the number of mitochondria within each axonal bundle at minute 1 and 5. Data were analysed by paired Student's *t* test. Number of wings analysed: wt-Miro = 5, Split-Miro = 6, from

2 independent experiments. **(G, H)** Anterograde velocity (G) and run length (H) of axonal mitochondria in wing neurons expressing wt-Miro or Split-Miro, during the first and fifth minute of blue light exposure, relative to (E, F). Circles represent tracked mitochondria. Data were analysed by Student's *t* test. Due to the overall lower number of bidirectional and retrograde-moving mitochondria, a meaningful statistical analysis of their velocity and run length is not possible. The data underlying the graphs shown in the figures can be found in S1 Data. (TIF)

**S6 Fig. The role of actin in the Split-Miro–dependent retraction of the mitochondrial network in S2R+ cells. (A)** S2R+ cell transfected with mCherry/EGFP-tagged Split-Miro and not treated with cytochalasin D (to maintain an intact actin network) were imaged by time-lapse with a 488-nm laser for 7 minutes. The presence of an intact actin network does not prevent mitochondrial network collapse. Scale bar: 10 μm. Cartoon depicts the reconstitution state of Split-Miro at 0 and 7 minutes (min) under blue light. **(B)** Representative images of S2R+ cells treated with or without cytochalasin D (cytoD) for 4 hours before imaging. Before imaging, cells were stained with MitoTracker Green (MitoTracker) and ActinTracker DeepRed (Actin-Tracker) to visualise the mitochondria and the actin network, respectively. Note the depolymerisation of the actin network after cytoD treatment, which is required for process extension in S2R+ cells. Scale bar: 5 μm. (TIF)

**S1 Table. Plasmids used in this study.** (DOCX)

**S2 Table. Primers used in this study.** For NEBuilder HiFi DNA Assembly, bold fonts indicate primer overlap with the amplified gene product, underline indicated overlap with the adjacent insert or plasmid sequence. For site-directed mutagenesis, bold fonts indicate mutated nucleotides. For dsRNA template production, underline indicates T7 promoter sequence which precedes a sequence overlapping with either Miro 3′-UTR (dsRNA) or a nontargeting sequence within the pT2-DsRed-UAS plasmid backbone (Control). (DOCX)

**S3 Table. Fly lines used in this study and genotypes analysed in the behavioural assay.** (DOCX)

**S1 Movie. Representative time-lapse movie of S2R+ cells transfected with mCherry-Split-MiroN (magenta) and EGFP-tagged Split-MiroC (green).** mCherry-Split-Miro-EGFP is reconstituted at the mitochondria, and, upon irradiation with 488-nm blue light, mCherry-Split-MiroN (but not EGFP-tagged Split-MiroC) is released into the cytoplasm and fully reconstitutes within 3 minutes. Note mCherry-Split-MiroN release is already evident at 2 seconds, followed by complete diffusion throughout the cytoplasm. (AVI)

**S2 Movie. Representative time-lapse movie of S2R+ cells transfected with mCherry-Split-MiroN and untagged Split-MiroC, before and after a 570-ms pulse of blue light.** Upon irradiation with 488-nm blue light, mCherry-Split-MiroN (grey) is released from the mitochondria into the cytoplasm and fully reconstitutes within 120 seconds. (MOV)

**S3 Movie. Representative time-lapse movie of S2R+ cells transfected with mCherry-Split-MiroN and EGFP-tagged Split-MiroC and imaged with the 488-nm laser to capture EGFP-targeted mitochondria (grey) while photocleaving Split-Miro.** Note the strong

mitochondrial morphological changes during the period of blue light imaging. Not shown, mCherry-Split-MiroN.
(AVI)

**S4 Movie. Representative time-lapse movie of S2R+ cells transfected with mCherry-Split-MiroN and EGFP-tagged Split-MiroC and imaged with the 561-nm laser for 1 minute (to capture Split-Miro via mCherry, red) before switching to the 488-nm laser for 7 minutes to capture EGFP-targeted mitochondria (green) while photocleaving Split-Miro.** Note the strong mitochondrial morphological changes and reduced motility during the period of blue light imaging.
(AVI)

**S5 Movie. Representative time-lapse movie of S2R+ cells transfected with mCherry-wt-Miro and EGFP-tagged Split-MiroC and imaged with the 561-nm laser for 1 minute (to capture wt-Miro via mCherry, red) before switching to the 488-nm laser for 7 minutes to capture EGFP-targeted mitochondria (green).** Note that, contrarily to Split-Miro transfected cells (S4 Movie), there are no mitochondrial morphological changes or reduced motility during the period of blue light imaging.
(AVI)

**S6 Movie. Representative time-lapse movie of mitochondria motility in a 30-μm axonal tract of adult wing neurons expressing either wt-Miro (top movie) or Split-Miro (bottom movie) in a *miro*$^{+/-}$ background.** Mitochondrial motility is shown after 1 and 5 minutes of blue light exposure. Each movie corresponds to 30 seconds of real-time imaging.
(AVI)

## Acknowledgments

We thank Tito Calì, Manolis Fanto, Gohta Goshima, Marc-David Ruepp, Tom Schwarz, Zu-Hang Sheng, Konrad Zinsmaier, and Alex Whitworth for sharing reagents, the Fly Facility of the Department of Genetics, University of Cambridge for help with *Drosophila* embryo injections, the BFK Lab for assistance with the behavioural assays, the Wohl Cellular Imaging Centre at King's College London for help with light microscopy, and the Bloomington Drosophila Stock Center for fly stocks. We thank members of the Vagnoni lab, Joe Bateman and Simon Bullock for critically reading the manuscript.

## Author Contributions

**Conceptualization:** Francesca Mattedi, Alessio Vagnoni.

**Data curation:** Francesca Mattedi, Ethlyn Lloyd-Morris, Alessio Vagnoni.

**Formal analysis:** Francesca Mattedi, Ethlyn Lloyd-Morris, Alessio Vagnoni.

**Funding acquisition:** Frank Hirth, Alessio Vagnoni.

**Investigation:** Francesca Mattedi, Ethlyn Lloyd-Morris, Alessio Vagnoni.

**Methodology:** Francesca Mattedi, Ethlyn Lloyd-Morris, Alessio Vagnoni.

**Project administration:** Alessio Vagnoni.

**Resources:** Frank Hirth, Alessio Vagnoni.

**Supervision:** Alessio Vagnoni.

**Validation:** Francesca Mattedi, Ethlyn Lloyd-Morris, Alessio Vagnoni.

**Visualization:** Francesca Mattedi, Ethlyn Lloyd-Morris, Alessio Vagnoni.

**Writing – original draft:** Alessio Vagnoni.

**Writing – review & editing:** Francesca Mattedi, Frank Hirth, Alessio Vagnoni.

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
