## [Editor Report · Decision Letter 0]

23 Oct 2022

Dear Dr Vagnoni, 

Thank you for submitting your manuscript entitled "Optogenetic Miro cleavage reveals direct consequences of real-time loss of function in Drosophila" for consideration as a Research Article by PLOS Biology.

Your manuscript has now been evaluated by the PLOS Biology editorial staff as well as by an academic editor with relevant expertise and I am writing to let you know that we would like to send your submission out for external peer review.

Once your full submission is complete, your paper will undergo a series of checks in preparation for peer review. After your manuscript has passed the checks it will be sent out for review. To provide the metadata for your submission, please Login to Editorial Manager (https://www.editorialmanager.com/pbiology) within two working days, i.e. by Oct 25 2022 11:59PM.

Kind regards,

Ines

--

Ines Alvarez-Garcia, PhD

Senior Editor

PLOS Biology

---

## [Decision Letter · Decision Letter 1]

23 Dec 2022

Dear Dr Vagnoni,

Thank you for your patience while your manuscript entitled "Optogenetic Miro cleavage reveals direct consequences of real-time loss of function in Drosophila" was peer-reviewed at PLOS Biology. Please also accept my apologies for the delay in providing you with our decision. The manuscript has now been evaluated by the PLOS Biology editors, an Academic Editor with relevant expertise, and by three independent reviewers. 

The reviews are attached below. As you will see, the reviewers find the results potentially interesting and the tools used powerful, however they are also underwhelmed by the depth of the characterisation of the system and by the description of the role of Split-Miro in vivo. They think more experiments should be performed to analyse further the impact of Miro cleavage on mitochondrial interactions with the ER and the role of Miro in animal physiology. In addition, Reviewer 2 is not convinced by the experiments showing the dissociation and reassociation of Split-Miro in response to blue light, whether or not some Miro functions are retained after light-induced dissociation, how efficient is the dissociation or if there is a dominant negative effect after dissociation. After discussing the reviews with the Academic Editor, we would like to invite you to revise the work to thoroughly address the reviewers' concerns.

Given the extent of revision needed, we cannot make a decision about publication until we have seen the revised manuscript and your response to the reviewers' comments. Your revised manuscript is likely to be sent for further evaluation by all or a subset of the reviewers.

**IMPORTANT - SUBMITTING YOUR REVISION**

3. Resubmission Checklist

a) *PLOS Data Policy*

b) *Published Peer Review*

Sincerely,

Ines

--

Ines Alvarez-Garcia, PhD

Senior Editor

PLOS Biology

Reviewers' comments

Rev. 1:

This manuscript by Mattedi et al. describes the use of optogenetic manipulation of the Miro GTPase in Drosophila. This is a challenging but potentially very useful approach to study the acute effects of loss of a functional Miro in an animal model. Most of the studies were conducted in insect cells but the study included an effort also to analyze the activity of adult flies. I find the manuscript very interesting, the experiments have been carried out in an appropriate manner and the text is well written and follows a logical flow. I only have some minor comments:

1. The design of the split-Miro construct result in a loss of mitochondrial targeting of Miro since the LOV2:Zdk1 pair is inserted between the C-terminal GTPase domain and the transmembrane domain of Miro. Blue light releases the major part of Miro from the transmembrane domain; however, I believe the GTPase domain and EF hands are still intact and active. If I understand correctly, the result is therefore not really a loss of function but rather a loss of mitochondrial targeting. It is clear that the mislocalization negatively affects the mitochondrial motility but could there be any adverse effects by the released Miro deletion mutant? The released Miro variant could maybe sequester Miro binding proteins in the cytoplasm and the effects might not be apparent on the time-scale examined in the experimental design. What is the view of the authors on this possibility?

2. I am a bit surprised by the increased activity of the adult flies described in Fig.5. It is not entirely clear what is meant by "activity" for a non-fly expert. What precisely is measured? I thought that a decreased mitochondrial motility would result in decreased neuro-muscular function and hence reduced activity. Why do we see the opposite in this experimental set up? The authors could elaborate a bit more on this finding.

3. Please remove the sentence "To our knowledge, this is the first example of tunable animal behavior by real-time loss of protein function" (lines 68-69). This statement is irrelevant.

Rev. 2:

The study by Mattedi et al developed an optogenetic 'Split-Miro' system controlling Miro-dependent mitochondrial functions in Drosophila that is based on the light-oxygen-voltage-sensing domain 2 (LOV2) and its binding partner ZDark1 (zdk1). The feasibility of the Split-Miro system is demonstrated in Drosophila S2R+ cells. Acute dissociation of Split-Miro in S2R+ cells disrupts the processivity of microtubules-based mitochondrial transport and causes a rearrangement of the mitochondrial network, as it has been previously observed for loss of function mutations. However, dissociation of Split-Miro had no effect on the ability of mitochondria to buffer calcium in contrast to previous studies. The study further shows the Split-Miro protein is at least partially functional in intact animals that acute inactivation of Miro function causes an unexplained hyperactivity in flies.

1) The experiment showing the dissociation and reassociation of Split-Miro is not convincing (Fig 1D-F). The representative image shows a diming of mitochondria-associated mCherry-Split-MiroN but not the expected increase of the mCherry signal in the cytosol. I am also confused why only MiroN but not MiroC is tagged in this experiment.

Since it is critical to demonstrate how well the Split-Miro system dissociates in response to blue light and reassociates in its absence, I suggest modifying the experiment by using a tagged (instead of untagged) Split-MiroC version. This will allow a quantification of a) how much mCherry-Split-MiroN becomes cytosolic and b) how much mCherry-Split-MiroN remains associated with mitochondria in the presence of blue light. The same principle should be applied to reassociation.

2) Fig 1C. The shown reconstitution of Split-Miro at mitochondria in the absence of blue light requires quantification to clearly demonstrate the efficiency of the Split-Miro system.

3) The age-dependent effects of acute Split-Miro inactivation increasing locomotor activity are interesting but unexplained. Any follow-up experiment providing some evidence to explain this effect will significantly strengthen the manuscript.

4) The suggested model for Miro-mediated regulation of mitochondrial motility is not new and should be eliminated from the manuscript. Previously, Glater et al (2006, JCB) showed that a truncated Milton protein containing residues 847-1,116 can still associate with mitochondria even though Miro binding is abolished. Therefore, Glater et al hypothesized Milton-motor complexes can also bind to an

unidentified mitochondrial protein in a Miro-independent manner.

5) In the discussion section the authors hypothesize that the impairment of mitochondrial Ca2+ homeostasis that has been demonstrated with Miro loss of function approaches may be a secondary effect of Miro loss of function. While this is reasonable, the authors should also mention the possibility that this presumed and well documented function of Miro could be linked to the TM domain or the few amino acids of the C-terminus of Miro, which are known to interact with Sam and MICOs complexes. Accordingly, the remaining Split-MiroC may still mediate mitochondrial Ca2+ homeostasis in the presence of blue light.

Minor

Fig 1a: The Figure is drawn as if a lov2-ZDK1 cassette is being inserted into Miro, which is confusing. I suggest showing the two generated constructs (MiroN and -C) independently.

Rev. 3:

This is a potentially interesting manuscript that focuses on the dynamic role of Miro GTPases for regulating mitochondrial trafficking. Miro proteins have long been know to play important roles in regulating mitochondrial distribution and morphology by coupling mitochondria to microtubule dependent transport. Most studies to date have used either loss or gain of function approaches in a wide variety of in vitro and in vivo model systems including worms, flies, mice and human cells. However the implications of acute disruption of Miro on mitochondrial dynamics are less well understood. To address this the authors have used an optogenetic technique allowing for reversible light based disruption of drosophila Miro function by taking advantage of the LOV2 system whereby a LOV2 domain is fused to the C-terminus of the (N-terminal) cytoplasmic part of dMiro while the Miro TM has an N-terminal ZDK1 domain. In the dark state these two domains reconstitute a protein complex (apparently making a functional Miro) that can then be disrupted by green light and hence allowing for rapid optogenetic Miro 'photocleavage'. On the whole the experiments are well performed throughout.

While the approach is elegant and has elements of state of the art and there are a number of potentially interesting findings, enthusiasm is dampened by the lack of major new mechanistic insight regarding Miro function gleaned from the approach. The authors do not seem to have gone into much depth with their model leaving many issues unaddressed despite the potential of the system. The contribution to advancing understanding Miro function therefore remains quite superficial.

The authors focus on microtubule dependent transport but Miro has also been proposed to link mitochondria to Myosin motors. What are the consequences of acute Miro disruption for actin and myo19-dependent roles of Miro and mitochondrial dynamics (e.g. in vivo).

Quantification (according to methods) reveals the extent of Miro release into the cytoplasm. The data in Figure 1 therefore needs further quantification e.g. of the loss/gain of Miro N-term fluorescence on the mitochondria (presented as Fig1A-C). It seems important to know the dynamics of Miro remaining on the mitochondria upon phtocleavage - not just what is cytoplasmically released.

It's somewhat unclear what is novel in Figure 2. This seems to be mostly over expression data and numerous studies have already looked at the impact of mitochondrial trafficking of Miro up regulation. Moreover, it is unclear if this is a physiological relevant cell type to add significant insight. While the data do provide further support that split Miro behaves in a similar way to Miro it does not provide much new insight on the importance of miro for regulating mitochondrial trafficking.

Can the authors present examples of the presumably highly artificial 'processes' induced by cytochalasin D treatment? These processes apparently have microtubules of uniform polarity (e.g. plus end out) but they require actin depolymerisation for their formation. How does this impact interpretation given Miro also couples to actin transport pathways.

The speed at which mitochondrial rearrangement occurs is is certainly intriguing but the overall phenotype of light induced Miro loss of function is very reminiscent of what has been shown previously upon Miro loss of function or upon calcium-dependent mitochondrial uncoupling from microtubules. ie collapsed mitochondrial networks with shorter mitochondria. The authors do not sufficiently articulate what major novel finding they achieve using the new optogenetic approach.

Moreover from the movies it looks like large amounts of trafficking remain upon photocleavage? Can the authors explain this? What underlies remaining movement?? Is this microtubule dependent?

The authors mention that there is no effect on peroxisome distribution but what about trafficking kinetics of these organelles?

It is also somewhat surprising that the authors do not further investigate the impact of Miro cleavage on mitochondrial interactions with the ER. it seems a significantly missed opportunity to explore these aspects given that much of what the authors demonstrate with split miro is confirmatory of what is already known about Miro function.

Although the rescue by syntaphalin is potentially interesting several of the images in Fig 3 e.g. E, E' are of poor resolution and make it hard to see what is presented in the qualified data.

The authors demonstrate that Miro loss of function does not disrupt calcium entry into mitochondria. This does not seem particularly surprising given than the main calcium entry pathway is through MCU on the IMM. Miro has been shown to form complexes with MCU but at least in mammalian cells mitochondria lacking Miro can still take up calcium.

The in vivo experiments provide potential interesting findings supporting an important role for Miro in animal physiology. But there is currently insufficient data reporting the impact of Miro cleavage on the mitochondrial network in vivo or what the cellular consequences. As such it remains unclear how photocleavage leads to the altered physiology.

Movies: The authors should provide a similar example movie imaging mitochondria under identical photo illumination conditions in cells not over expressing Miro or expressing WT-Miro to demonstrate that the illumination conditions used to disrupt split miro do not impact mitochondrial trafficking.

The example movies provided (e.g. Figure 2) reveal that upon photo cleavage parts of the mitochondrial network remodel. However it also clear that large amounts of bi-directional trafficking of mitochondria remain throughout the movie.

---

## [Decision Letter · Decision Letter 2]

28 Jun 2023

Dear Dr Vagnoni,

Thank you for your patience while we considered your revised manuscript entitled "Optogenetic Miro cleavage reveals direct consequences of real-time loss of function in Drosophila" for publication as a Research Article at PLOS Biology. This revised version of your manuscript has been evaluated by the PLOS Biology editors, the Academic Editor and one of the original reviewers.

Based on the review and our Academic Editor's assessment of your revision, we are likely to accept this manuscript for publication, provided you satisfactorily address the data and other policy-related requests stated below.

In addition, we would like you to consider a suggestion to improve the title:

"Optogenetic spatio-temporal control of GTPase Miro activity reveals its role in mitochondrial network maintenance and controlling locomotor activity in adult Drosophila"

We expect to receive your revised manuscript within two weeks. 

*Published Peer Review History*

*Press*

Sincerely,

Ines

--

Ines Alvarez-Garcia, PhD

Senior Editor

PLOS Biology

DATA POLICY:

Many thanks for providing the data underling all the graphs shown in the figures. Please also ensure that figure legends in your manuscript include information on where the underlying data can be found - for example, you can add in all the corresponding figure legends (both main and supplementary) the following: "The data underlying the graphs shown in the figure can be found in Data S1."

We require the original, uncropped and minimally adjusted images supporting all blot and gel results reported in an article's figures or Supporting Information files. We will require these files before a manuscript can be accepted so please prepare and upload them now. Please carefully read our guidelines for how to prepare and upload this data: https://journals.plos.org/plosbiology/s/figures#loc-blot-and-gel-reporting-requirements

Reviewers' comments

Rev. 2:

The authors have appropriately addressed all my previous concerns.

---

## [Editor Report · Decision Letter 3]

22 Jul 2023

Dear Dr Vagnoni,

Thank you for the submission of your revised Research Article entitled "Optogenetic cleavage of the Miro GTPase reveals the direct consequences of real-time loss of function in Drosophila" for publication in PLOS Biology. On behalf of my colleagues and the Academic Editor, Anna Akhmanova, I am delighted to let you know that we can in principle accept your manuscript for publication, provided you address any remaining formatting and reporting issues. These will be detailed in an email you should receive within 2-3 business days from our colleagues in the journal operations team; no action is required from you until then. Please note that we will not be able to formally accept your manuscript and schedule it for publication until you have completed any requested changes.

PRESS

Sincerely, 

Ines

--

Ines Alvarez-Garcia, PhD

Senior Editor

PLOS Biology
